# Probabilistic Tensor Decomposition of Neural Population Spiking Activity

**Hugo Soulat**
Gatsby Unit
University College London
London, W1T 4JG
`hugos@gatsby.ucl.ac.uk`

**Sepiedeh Keshavarzi**
Sainsbury Wellcome Centre
University College London
London, W1T 4JG
`s.keshavarzi@ucl.ac.uk`

**Troy W. Margrie**
Sainsbury Wellcome Centre
University College London
London, W1T 4JG
`t.margrie@ucl.ac.uk`

**Maneesh Sahani**
Gatsby Unit
University College London
London, W1T 4JG
`maneesh@gatsby.ucl.ac.uk`

## Abstract

The firing of neural populations is coordinated across cells, in time, and across experimental conditions or repeated experimental trials, and so a full understanding of the computational significance of neural responses must be based on a separation of these different contributions to structured activity. Tensor decomposition is an approach to untangling the influence of multiple factors in data that is common in many fields. However, despite some recent interest in neuroscience, wider applicability of the approach is hampered by the lack of a full probabilistic treatment allowing principled inference of a decomposition from non-Gaussian spike-count data. Here, we extend the Pólya-Gamma (PG) augmentation, previously used in sampling-based Bayesian inference, to implement scalable variational inference in non-conjugate spike-count models. Using this new approach, we develop techniques related to automatic relevance determination to infer the most appropriate tensor rank, as well as to incorporate priors based on known brain anatomy such as the segregation of cell response properties by brain area. We apply the model to neural recordings taken under conditions of visual-vestibular sensory integration, revealing how the encoding of self- and visual-motion signals is modulated by the sensory information available to the animal.

## 1 Introduction

Large-scale neural population recording offers a unique window through which to study brain computations that involve the coordinated action of many neurons. Although such rich data sets are increasingly common, the information they contain can only be put to full use once we are able to separate the underlying factors that contribute to the neural activity in a simple and interpretable way. Rich data sets often fit naturally within a multidimensional array or tensor, and a variety of tensor decomposition techniques are available to identify factorial contributions that shape such data. But whereas tensor decompositions have long been used in fields such as chemometrics [1] or computer vision [2], until recently their use in neuroscience had been limited to the study of continuous traces like EEG [3], fMRI signals [4] or LFP [5]. Encouragingly, in the last few years, tensor methods have proven useful to segregate the influence of time dynamics, trial to trial variability [6, 7] or

35th Conference on Neural Information Processing Systems (NeurIPS 2021).

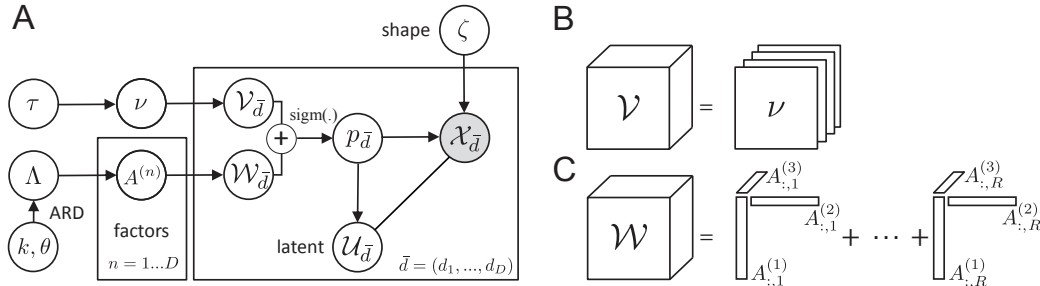

Figure 1: Probabilistic tensor decomposition model. (A) Graphical model. (B) The Offset tensor $\mathcal{V}$ is constrained to vary along a limited set of dimensions. (C) $\mathcal{W}$ is a low rank tensor.

experimental condition [8] in neural spike-count data, even though the models employed have not yet provided a full probabilistic treatment appropriate to counts.

Canonical Polyadic (CP) decomposition [9] is a widely used technique in which a tensor is decomposed into a sum of rank-1 elements. When applied to noisy data, it is usually cast as a least-squares problem and therefore implicitly assumes that noise is normally distributed. Spike counts violate this assumption but alternated-gradient-based optimization methods can be adapted to non-Gaussian likelihood functions. Hong et al. [10] introduced a general framework for Generalized CP (GCP) decomposition that can be applied to Poisson or negative binomial (NB) distributed datasets but it has not been used in neuroscience to our knowledge. Moreover, practical methods have largely focused on point estimates of the decomposition rather than a full probabilistic treatment. As such, they do not provide principled ways to incorporate prior knowledge about the data (such as the (co)location of recorded neurons in the brain), to automatically determine the rank of the observed tensor (although it is a generalization of the matrix rank, tensor rank is not as well behaved and understood [9]), nor to estimate posterior confidence in results.

One approach to the Bayesian treatment of count models is based on augmentation using Pólya-Gamma (PG) variables [11]. This solution enables Gibbs sampling in models which combine Gaussian latent structure with logistic, Poisson or negative binomial observations, including factor models [12] and generalised-linear regression [11], and it has been used successfully in neuroscience settings [13]. While PG augmentation has also been considered for tensor decomposition [14], these authors recognised the computational challenges posed by Gibbs sampling in typical size problems, and so focused primarily on point estimation while incorporating a limited range of priors.

Here, we introduce a variational approach to PG-augmented models with negative-binomial observations, which enables a relatively efficient Variational Bayesian (VB) treatment. Our approach is based on a principled approximation of the PG cross entropy, which is especially well suited for neural datasets in which conditional Fano factors (FF) (once the contribution of population-wide shared influences is discounted) are often close to one [15, 16]. The fully probabilistic formulation is able to handle missing observations, extending to situations where data from multiple animals or recording sessions are to be combined (sometimes referred to as "stitching" [17]). VB also makes it possible to combine Automatic Relevance Determination (ARD) with knowledge about expected group structures based on anatomical or histological information about the recorded neurons (brain area, morphology, protein expression etc.). Finally, we also augment standard CP decomposition models using a constrained offset tensor, which allows us to study modulation of neural activity around baseline values, improving readability and interpretability of the CP-factors.

The paper is organised as follows: In section 2 we review background material on tensor decomposition and PG augmentation schemes. In section 3, we derive a VB algorithm for approximate inference of the tensor factors and model parameters. Last, in section 4, we analyse synthetic data and neural recordings in mice performing passive multisensory integration [18] and show that our method can estimate the population-level effects of temporal dynamics and experimental condition in a fully probabilistic manner. We show improvements in variance explained, deviance and, last but not least, decomposition robustness when compared to standard CP and GCP baselines.

## 2   Background

### 2.1   Tensor decomposition

Consider observed spike counts gathered in a $D$-dimensional tensor

$$\mathcal{X} \in \mathbb{R}^{I_1 \times \cdots \times I_D} \,, \tag{1}$$

with dimensions corresponding to neurons, time in trial, various (factorial) experimental manipulations, and possibly trial number. Our goal is to decompose $\mathcal{X}$ into a pair of simpler objects: a low rank tensor $\mathcal{W}$ and an "offset" tensor $\mathcal{V}$ which varies along a restricted set of dimensions. Here $\mathcal{W}$ is designed to capture low-dimensional structure in the time-course of population activity that varies systematically with experimental condition, while the offset $\mathcal{V}$ models potential changes in baseline firing rates with trial or experimental condition.

More specifically, we seek a rank-$R$ Canonical-Polyadic (CP) [9] decomposition of $\mathcal{W}$, often denoted $\mathcal{W} = [\![A^{(1)}, \ldots, A^{(D)}]\!]$, such that for $\bar{d} = (d_1, \ldots, d_D) \in \prod_{d=1}^{D}\{1 \ldots I_d\}$:

$$\mathcal{W}_{\bar{d}} = \sum_{r=1}^{R} A_{d_1 r}^{(1)} \ldots A_{d_D r}^{(D)} \,. \tag{2}$$

For $n = 1 \ldots D$, $A^{(n)}$ is an $I_n \times R$ matrix, termed a factor, whose rows are $a_i^{(n)} = [A_{i1}^{(n)}, \ldots, A_{iR}^{(n)}]$. For a tensor $\mathcal{T}$, we represent by $\mathcal{T}_{(n)}$ its $n$-th unfolding and recall that:

$$\mathcal{W}_{(n)} = A^{(n)} B^{(n)\mathsf{T}} \,, \tag{3}$$

where $B^{(n)}$ is the Khatri-Rao product $\bigodot_{p \neq n} A^{(p)}$.

In addition, we assume that $\mathcal{V}$ varies only along a subset of $D_* \leq D$ dimensions, remaining fixed along the complementary dimensions. For convenience, in what follows we will write $\bar{d} = \bar{d}^* \cup \bar{d}^\bullet$ where $\bar{d}^* = (d_1^*, \ldots, d_{D_*}^*)$ contains the dimesions that vary and $\bar{d}^\bullet = (d_1^\bullet, \ldots, d_{D-D_*}^\bullet)$ those that are fixed, regardless of the dimension ordering. Thus, instead of a $I_1 \times \cdots \times I_D$ tensor, in practice we need only estimate the $I_1 \times \cdots \times I_{D_*}$ tensor $\nu$ and define $\mathcal{V}$ such that for all $\bar{d}^*, \bar{d}^\bullet$:

$$\mathcal{V}_{\bar{d}^* \cup \bar{d}^\bullet} = \nu_{\bar{d}^*} \,. \tag{4}$$

We then take $\mathcal{X}$ to be generated by a negative-binomial noise process from the decomposition determined by $\mathcal{W}$ and $\mathcal{V}$.

### 2.2   Negative-binomial distribution

The negative binomial (NB) model can be seen as a doubly stochastic relaxation of the Poisson distribution to overdispersed (FF $>$1) data. We model each spike count $\mathcal{X}_{\bar{d}}$ as Poisson-distributed with random mean $\mu_{\bar{d}}$ drawn from a Gamma distribution with shape $\zeta$ and scale $e^{\mathcal{W}_{\bar{d}} + \mathcal{V}_{\bar{d}}}$:

$$\begin{aligned} (\mathcal{X}_{\bar{d}} | \mu_{\bar{d}}) &\sim \text{Poisson}(\mu_{\bar{d}}) \,, \\ (\mu_{\bar{d}} | \mathcal{V}_{\bar{d}}, \mathcal{W}_{\bar{d}}, \zeta) &\sim \text{Gamma}(\zeta, e^{\mathcal{W}_{\bar{d}} + \mathcal{V}_{\bar{d}}}) \,. \end{aligned} \tag{5}$$

Marginalizing over the Poisson rates $\mu_{\bar{d}}$ then yields the NB distribution [13] :

$$P(\mathcal{X}_{\bar{d}} | \mathcal{V}_{\bar{d}}, \mathcal{W}_{\bar{d}}, \zeta) = \frac{\Gamma(\zeta + \mathcal{X}_{\bar{d}})}{\mathcal{X}_{\bar{d}}! \, \Gamma(\zeta)} (1 - p_{\bar{d}})^{\zeta} (p_{\bar{d}})^{\mathcal{X}_{\bar{d}}} \quad \Rightarrow \quad (\mathcal{X}_{\bar{d}} | \mathcal{V}_{\bar{d}}, \mathcal{W}_{\bar{d}}, \zeta) \sim \text{NB}(\zeta, p_{\bar{d}}) \,, \tag{6}$$

where $p_{\bar{d}} = 1/(1 + e^{-(\mathcal{W}_{\bar{d}} + \mathcal{V}_{\bar{d}})})$ is the success probability. We recall that the mean and variance of the NB distribution are respectively given by $\mathbb{E}(\mathcal{X} | \zeta, \mathcal{W}, \mathcal{V}) = \zeta e^{\mathcal{W} + \mathcal{V}}$ and $\mathbb{V}(\mathcal{X} | \zeta, \mathcal{W}, \mathcal{V}) = \zeta e^{\mathcal{W} + \mathcal{V}}(1 + e^{\mathcal{W} + \mathcal{V}})$, giving a Fano factor FF $= 1 + e^{\mathcal{W}_{\bar{d}} + \mathcal{V}_{\bar{d}}}$ that is greater than one and approaches the Poisson case when $\zeta$ increases to infinity while keeping the mean constant.

### 2.3   Pólya-Gamma augmentation

Unfortunately, the negative binomial observation probability does not admit a conjugate prior on $\mathcal{W}$ and $\mathcal{V}$ which makes direct Bayesian inference intractable. This challenge is resolved by adopting a PG augmentation [11]. We first recall key properties of PG distributions before deriving the augmented model likelihood.

### 2.3.1 Definition and properties

For $\xi \in \mathbb{R}^+$ and $\omega \in \mathbb{R}$, a random variable $U$ is $\text{PG}(\xi, \omega)$ distributed iff:

$$U \stackrel{D}{=} \frac{1}{2\pi^2} \sum_{k=1}^{\infty} \frac{g_k}{(k-1/2)^2 + \omega^2/(4\pi^2)} , \tag{7}$$

where $g_k \sim \text{Gamma}(\xi, 1)$. We recall three major results from [11].

**(i)** For $\xi > 0$, if $U \sim \text{PG}(\xi, 0)$, then for all $\zeta, v$:

$$\frac{e^{v\zeta}}{(1+e^v)^\xi} = 2^{-\xi} e^{(\zeta - \xi/2)v} \int_0^\infty e^{-\frac{U}{2}v^2} p(U|\xi, 0) dU . \tag{8}$$

**(ii)** The densities of $\text{PG}(\xi, \omega)$ and $\text{PG}(\xi, 0)$ are linked through:

$$p(U|\xi, \omega) = \cosh^\xi(\omega/2) e^{-\frac{U}{2}\omega^2} p(U|\xi, 0) . \tag{9}$$

**(iii)** Although the PG density can be expressed as an alternating-sign sum of inverse gamma distributions, the terms of the sum diverge rapidly as $\xi$ increases. However, the Laplace transform of the density (the sign-reflected moment generating function) has the closed form:

$$\mathbb{E}\left(e^{-tU}\right) = \cosh^\xi(\omega/2) \cosh^{-\xi}\left(\sqrt{\frac{\omega^2/2 + t}{2}}\right) . \tag{10}$$

Moments of the distribution are obtained from derivatives of this transform at 0, giving:

$$\mathbb{E}(U) = \frac{\xi}{2\omega} \tanh(\omega/2) \quad \text{and} \quad \mathbb{V}(U) = \frac{\xi}{4\omega^3 \cosh^2(\omega/2)} \left(\sinh(\omega) - \omega\right) , \tag{11}$$

with similar closed-form expressions for higher-order moments.

These properties will, respectively, allow us to (i) augment our model with a latent tensor and obtain a Gaussian likelihood in $\mathcal{W}$ and $\mathcal{V}$, (ii) derive the latent posterior distribution and (iii) introduce and validate a moment-matching approximation to PG cross entropies, thus facilitating variational Bayesian inference.

### 2.3.2 Model augmentation

Let $x \sim \text{NB}(\zeta, p)$, with $p = 1/(1 + e^{-v})$. Introducing a random variable $u \sim \text{PG}(\zeta + x, 0)$ we combine (6) and (8) to obtain

$$P(x|\zeta, v) = \frac{\Gamma(\zeta + x)}{x! \, \Gamma(\zeta)} \frac{(e^{-v})^\zeta}{(1 + e^{-v})^{\zeta + x}} = \frac{\Gamma(\zeta + x)}{x \Gamma(\zeta)} 2^{-(\zeta + x)} e^{\left(\frac{x - \zeta}{2}\right)v} \int_0^\infty e^{-\frac{u}{2}v^2} p(u|\zeta + x, 0) du . \tag{12}$$

This expression allows us to condition on $u$ to obtain the augmented likelihood:

$$P(x|\zeta, u, v) = F(x, \zeta) e^{-\frac{u}{2}\left(v - \frac{x - \zeta}{2u}\right)^2} \tag{13}$$

(with $F$ gathering terms independent of $u$ and $v$) which is conjugate to a Gaussian prior on $v$.

Applying this augmentation to the tensor model, we introduce a tensor of PG variates $\mathcal{U}$ of the same size as $\mathcal{X}$, with each $\mathcal{U}_{\bar{d}} \sim \text{PG}(\zeta + \mathcal{X}_{\bar{d}}, 0)$ to obtain the log-likelihood

$$\log P(\mathcal{X}|\mathcal{U}, \mathcal{V}, \mathcal{W}, \zeta) =_{+C} -\frac{1}{2} \sum_{\bar{d}} \mathcal{U}_{\bar{d}} \left(\mathcal{W}_{\bar{d}} + \mathcal{V}_{\bar{d}} - \frac{\mathcal{X}_{\bar{d}} - \zeta}{2\mathcal{U}_{\bar{d}}}\right)^2 . \tag{14}$$

## 3 Structured variational inference

PG-based augmentation was developed to facilitate Gibbs sampling in otherwise non-conjugate models. However, despite the availability of closed-form updates, sampling quickly becomes impractical

for large datasets [14]. Thus, in this section, we propose a variational Bayes estimation procedure and report the associated updates (derived in Supplementary Materials A) to fit variational posteriors over the low rank tensor, the constrained offset tensor, and the tensor of PG latents. We incorporate prior knowledge about the neural recordings and perform Automatic Relevance Determination (ARD) using Gamma priors on the factor precision matrices. Last, we treat the shape $\zeta$ as a model parameter that we efficiently optimize during the variational M-step by approximating PG cross entropy terms with Gamma moment matching.

### 3.1  Priors

The prior on $\mathcal{U}$ is defined implicitly by the model augmentation through the conditional $P(\mathcal{U}_{\bar{d}}|\mathcal{X}_{\bar{d}}) = \mathrm{PG}(\zeta + \mathcal{X}_{\bar{d}}, 0)$. We set the prior on the offset matrix to be element-wise normal: $P(\nu_{\bar{d}*}) = \mathcal{N}(\mu_{\bar{d}*}, 1/\tau_{\bar{d}*})$, for $\bar{d}^* = (d_1^*, \ldots, d_{D_*}^*)$. Finally, for $n = 1 \ldots D$, $i = 1 \ldots I_n$ we use diagonal precision matrices $\Lambda_i^{(n)}$ to define the priors on the factor rows within $\mathcal{W}$:

$$P(a_i^{(n)}|\Lambda_i^{(n)}) = \mathcal{N}(0, \left(\Lambda_i^{(n)}\right)^{-1}). \tag{15}$$

The parametrisation of $\Lambda_i^{(n)}$ can encode various forms of structure expected in the data, for example:

**Case 1: Rank ARD.**  We assume that precision matrices are shared across a set of modes and rows and use a gamma prior to perform automatic relevance determination. That is, we take:

$$\Lambda_i^{(n)} = \mathrm{Diag}(\lambda_1, \ldots, \lambda_R), \quad \text{where } P(\lambda_r) = \mathrm{Gamma}(k_{0\lambda}, \theta_{0\lambda}). \tag{16}$$

During the optimization procedure, the shared diagonal elements of $\Lambda_i^{(n)}$ may diverge, effectively reducing the rank of the tensor.

**Case 2: Neuron-group constraints.**  Alternatively, we can impose a mode-specific precision. For example we can group neurons based on their recording site and impose shared precision matrices across neurons in each group. If neuron factors are gathered in $A^{(n)}$, we denote by $g(i) \in \mathcal{G}$ the group of neuron $i$ and define:

$$\Lambda_i^{(n)} = \mathrm{Diag}(\lambda_{g(i),1}, \ldots, \lambda_{g(i),R}), \quad \text{where } P(\lambda_{g(i)r}) = \mathrm{Gamma}(k_{0\lambda}, \theta_{0\lambda}). \tag{17}$$

In this case, divergence of a subset of precisions during optimization may lead components linked to one or more groups of neurons to shrink away. Thus, the model will favour explanations of the activity of each group that use as few components as possible.

### 3.2  Variational distribution

For the variational distribution on the latents $\mathcal{Z} = \{\mathcal{U}, \mathcal{V}, \mathcal{W}, \Lambda\}$, we adopt the factorisation:

$$q(\mathcal{Z}) = q(\mathcal{U})q(\mathcal{V})q(\mathcal{W})q(\Lambda) = \prod_{\bar{d}} q(\mathcal{U}_{\bar{d}}) \prod_{\bar{d}*} q(\nu_{d*}) \prod_{n,i} q(a_i^{(n)})q(\Lambda|k_\lambda, \theta_\lambda), \tag{18}$$

and then iteratively maximize the VB free energy (or ELBO) $\mathcal{F}(\zeta) = \langle \log P(\mathcal{X}, \mathcal{Z}|\zeta) \rangle_q + \mathcal{H}[q]$. If $q_{\neg x}$ corresponds to the variational distribution with variable $x$ marginalised out, we recall that the variational E-step update has the form $q(x) \propto \exp\langle \log P(\mathcal{X}, \mathcal{Z}|\zeta) \rangle_{q_{\neg x}}$ [19] (when obvious, we drop the explicit form of the distribution from the angle brackets).

### 3.3  Variational E-step

Here we describe the updates to infer the variational distributions of $\mathcal{W}, \mathcal{V}$ and $\mathcal{U}$. Details are provided in Supplementary A. We write $\langle \mathcal{Y}_{\bar{d}} \rangle := (\mathcal{X}_{\bar{d}} - \zeta)/(2\langle \mathcal{U}_{\bar{d}} \rangle)$, retaining the expectation brackets in the notation to emphasise the dependence on the complementary variational distributions.

**Update of low rank tensor $\mathcal{W}$.**  For $n = 1 \ldots D$, $i = 1 \ldots I_n$:

$$q\left(a_i^{(n)\mathsf{T}}\right) = \mathcal{N}\left(m_i^{(n)}, \Sigma_i^{(n)}\right), \tag{19}$$

where the mean and variance are given by:

$$\Sigma_i^{(n)} = \left( \left\langle B^{(n)\mathsf{T}} \text{Diag}\left( \langle \mathcal{U}_{(n),i:} \rangle \right) B^{(n)} \right\rangle + \left\langle \Lambda_i^{(n)} \right\rangle \right)^{-1}, \tag{20}$$

$$m_i^{(n)} = \Sigma_i^{(n)} \left( \langle B^{(n)\mathsf{T}} \rangle \text{Diag}\left( \langle \mathcal{U}_{(n),i:} \rangle \right) \right) \left( \langle \mathcal{Y}_{(n),i:} \rangle - \langle \mathcal{V}_{(n),i:} \rangle \right). \tag{21}$$

**Update of factor precisions $\Lambda$.**  By conjugacy, the precision update takes the form:

$$q(\Lambda | k, \theta) \sim \text{Gamma}(k_\lambda, \theta_\lambda), \tag{22}$$

where the parameters $k_\lambda$ and $\theta_\lambda$ depend on the form of the prior (Rank ARD or Neuron-group constraint).

**Update of offset tensor $\mathcal{V}$.**  For $\bar{d}^* = (d_1^*, \ldots, d_{D_1}^*)$, we need to sum over constrained dimension to update $q(\nu_{\bar{d}*})$:

$$q(\nu_{\bar{d}*}) = \mathcal{N}\left( m_{\bar{d}*}, \Sigma_{\bar{d}*} \right), \tag{23}$$

with

$$\Sigma_{\bar{d}*} = \left( \tilde{\mathcal{U}}_{\bar{d}*} + \tau_{\bar{d}*} \right)^{-1}, \quad m_{\bar{d}*} = \Sigma_{\bar{d}*} \left( \tilde{\mathcal{U}}_{\bar{d}*} \tilde{\mathcal{Y}}_{\bar{d}*} + \tau_{\bar{d}*} \mu_{\bar{d}*} \right), \tag{24}$$

and

$$\tilde{\mathcal{U}}_{\bar{d}*} = \sum_{\bar{d}\bullet} \langle \mathcal{U}_{\bar{d}* \cup \bar{d}\bullet} \rangle, \quad \tilde{\mathcal{Y}}_{\bar{d}*} = \frac{\sum_{\bar{d}\bullet} \langle \mathcal{U}_{\bar{d}* \cup \bar{d}\bullet} \rangle \left( \langle \mathcal{Y}_{\bar{d}* \cup \bar{d}\bullet} \rangle - \langle \mathcal{W}_{\bar{d}* \cup \bar{d}\bullet} \rangle \right)}{\sum_{\bar{d}\bullet} \langle \mathcal{U}_{\bar{d}* \cup \bar{d}\bullet} \rangle}. \tag{25}$$

**Update of PG latent $\mathcal{U}$.**  For $\bar{d} = (d_1, \ldots, d_D)$, using (12) and (9), we deduce that the PG variational distribution takes the form

$$q(\mathcal{U}_{\bar{d}}) = \text{PG}\left( \zeta + \mathcal{X}_{\bar{d}}, \Omega_{\bar{d}} \right), \text{ where } \Omega_{\bar{d}} = \sqrt{\left\langle \left( \mathcal{W}_{\bar{d}} + \mathcal{V}_{\bar{d}} \right)^2 \right\rangle}. \tag{26}$$

### 3.4  M-step

Finally, we fix the variational distribution and maximize the free energy with respect to the shape parameter. The main computational challenge in this step is to estimate the values of PG cross entropies. As we need one such estimate for each entry in $\mathcal{X}$, both sampling methods and numerical approximations of PG densities prove too expensive. We therefore developed a moment-matching procedure (see Supplementary Materials B) to obtain an efficient approximation for the regime where the conditional Fano factor is near 1.

Denote by $m_{PG}^n$ and $m_G^n$ the $n^{th}$ moments respectively of $\text{PG}(\xi, \omega)$ and of the Gamma distribution with matched mean and variance. As we show in Supplementary Materials B, the learnt model introduces coupling between the parameters $\xi$ and $\omega$ and the inferred conditional Fano Factor, with the result that both parameters diverge as FF $\to 1$ but the ratio $\xi e^{-\omega}$ remains finite. Furthermore, provided that $\Omega = \mathcal{O}(|\mathcal{W} + \mathcal{V}|)$ (ie. the variational posterior variance of the estimated tensor is of the order of its squared mean), the leading order terms in the polynomial expansions of $m_{PG}^n$ and $m_G^n$ are both equal to $\left( \frac{\xi}{2\omega} \right)^n$ and so their higher-order moments are asymptotically equivalent in the limit FF $\to 1$. This motivates an approximation of the PG KL divergence by the divergence between corresponding moment-matched gamma distributions.

Empirically, we find that the approximation is most accurate for small values of KL (see Supplementary Fig. 7), but in practice posterior and prior PG distributions may be very different. Fortunately, we can exploit the analytic properties of the PG distribution (Eq. 9) to re-write the free energy in terms of KL divergences between PG distributions which differ only in their first parameter:

$$\mathcal{F}(\zeta) =_{+C} \sum_{\bar{d}} \log \Gamma(\mathcal{X}_{\bar{d}} + \zeta)/\Gamma(\zeta) - \zeta \left( \log 2 + \langle \mathcal{W}_{\bar{d}}/2 \rangle + \langle \mathcal{V}_{\bar{d}}/2 \rangle + \log \cosh\left( \Omega_{\bar{d}}/2 \right) \right)$$
$$- \sum_{\bar{d}} \text{KL}\left( PG(\mathcal{X}_{\bar{d}} + \hat{\zeta}, \Omega_{\bar{d}}) || PG(\mathcal{X}_{\bar{d}} + \zeta, \Omega_{\bar{d}}) \right), \tag{27}$$

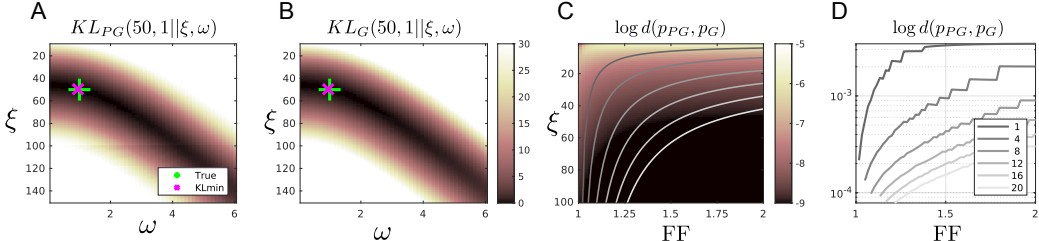

Figure 2: Moment-Matching Pólya-Gamma (PG) and Gamma (G) distribution in the unity Fano factor (FF) limit. (A) Numerical estimates of KL divergences between $PG(50, 1)$ and $PG(\xi,\omega)$. (B) Moment matched Gamma KL values. $KL_G(50, 1||\xi, \omega)$ indicates that we estimate $KL_{PG}(50, 1||\xi, \omega)$ using the closed formed KL divergence between two Gamma distribution moment matched to $PG(50, 1)$ and $PG(\xi, \omega)$. Color crosses point to the minimum of the KL divergences. (C) Normalized $L_2$ norm between the densities $p_{PG}$ and $p_G$ as a function of an effective Fano factor $FF = 1 + e^{-\omega}$, $d(f_{PG}, f_G) = 2||p_{PG} - p_G||^2_{L_2}/(||p_{PG}||^2_{L_2} + ||p_G||^2_{L_2}))$. The plain traces indicate relationship between $\xi$ and FF for different number of observed spikes. (D) Values of $d(f_{PG}, f_G)$ along such traces.

where we have omitted terms independent of $\zeta$, and $\hat{\zeta}$ is the current estimate of the shape parameter. The remaining KL divergence in (27) is comparatively small, and can be well approximated by the moment-matched gamma divergence.

We verified the validity of this approximation in numerical experiments (Supplementary B.5). The moment-matched gamma divergence replicated that between the PG distributions over a wide range (Figure 2 A-B) with accuracy increasing monotonically as conditional $FF \to 1$ (Figure 2 C-E).

It is important to stress that the quality of the approximation depends on the conditional rather than marginal Fano factor; that is the FF derived from residual variance after shared trial-to-trial variability captured by common factors in the population has been discounted. While marginal Fano factors that include all sources of variance in neural data are often found to exceed 1.5, population models tend to identify conditional Fano factors much closer to 1 [20]. Thus the moment-matched gamma approximation makes it possible to scale PG augmentation methods to large tensor decompositions in a regime appropriate to neural data analysis.

### 3.5  Dealing with missing data

Finally, we note that neural data analysis often faces missing data (failing electrode, confounded experiments, etc.) or multi-subject experiments, where a given neuron is only recorded in some trials. Although alternating gradient type updates, where the likelihood is maximized only over the observed entries, can accommodate such cases, they can be highly inefficient as they deal with a full size tensor. Instead, our variational scheme reduces the effective size of the observed dataset as the variational updates previously described are adjusted simply by considering observed entries only.

## 4  Experiments

In this section, we first explore the accuracy of tensor factorisations obtained by our proposed algorithm on simulated datasets. We confirm the robustness of the method to missing data, and compare the accuracy of ARD-based estimates of tensor rank to alternatives. We then benchmark the method against standard CP [9] and GCP [10] decompositions on neural spike data. We report performance in terms of variance explained, deviance explained, and a decomposition similarity metric. Finally, we look at the decomposition factors themselves, how they can be interpreted, and the consequences for neural data analysis.[1]

---

[1] All experiment were performed using an Intel(R) Xeon(R) CPU E5-2620 v3 @ 2.40GHz with 65GB of RAM. Benchmark analysis took approximately 12 hours.

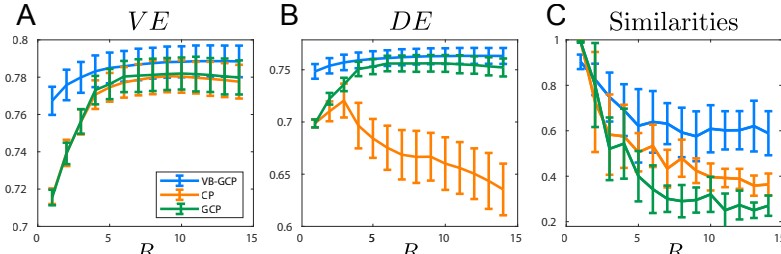

Figure 3: Held out (trial based) performances as a function of the tensor rank $R$ between our probabilistic model (VB-GCP), CP and Poisson GCP. (A) Variance Explained. (B) Poisson Deviance explained. (C) Similarity metric. Error bars indicate 1 standard deviation.

## 4.1 Simulations

We simulated two 5 dimensional data tensors (size $100 \times 70 \times 3 \times 5 \times 4$) using the generative NB model described in (6). In both cases, the generative shape parameter was fixed to $\zeta = 80$ and the generative tensor rank was $R = 4$. Assuming that the first dimension accounts for neuron loadings, we simulated "neuron groups" by restricting each CP-component to load only on a subset of neurons.

The first simulation included an offset tensor that was constrained to vary across the first and third dimension, and incorporated a multi-experiment stitching setting in which only 1/4 of the full tensor was observed. Supplementary Figures 8 and 9 show that the algorithm, initialized with $R = 6$ components, infers posteriors concentrated around the correct tensor decomposition, while identifying the correct shape parameter $\zeta$. Automatic relevance determination eliminates two of the components thus effectively estimating the rank of the data tensor. For comparison, using the exact same initialization but without ARD, two spurious components are identified (See Supplementary Figure 9).

The second simulation was designed to compare the quality of ARD-based rank estimates to other empirical rank selection metrics. To ensure standard CP and GCP methods [7, 9] were applicable we did not include an offset tensor. Results in Supplementary Figure 10 illustrate the fragility of alternative approaches.

## 4.2 Spike recordings

We then applied our method to neural spiking data [18] recorded in mice during a multisensory integration paradigm that aimed to elucidate the contribution of vestibular and visual signals to self-motion representation in the cortex. In brief, high-density single-unit recordings (Neuropixels and Neuronexus) were made from the retrosplenial cortex (RSP) of adult mice while they were presented with three types of motion stimuli: (i) passive horizontal rotation of the mouse in the dark (vestibular), (ii) passive horizontal rotation of the mouse in presence of a stationary surround vertical grating (vestibular + visual = both), and (iii) horizontal rotation of the surround vertical grating while the mouse was stationary, thus simulating optic flow alone (visual). (Figure 4-A). Single units were isolated in both granular and dysgranular divisions of the RSP (RSPg, RSPd) and across all cortical layers (Figure 4-C, left). The motion profile of the rotation is shown in Figure 4-B and was adapted such that the visual motion component under 'both' and 'visual' conditions were identical. The dataset consists of 676 neurons recorded under all experimental conditions over 10 trials. Each trial lasted 7 seconds (arranged in 0.1 second non overlapping time bins) during which the mouse was rotated back and forth from a reference 0 degree position to an angle of $\pi$. For each of the 24 cross validation folds, we split the dataset in half across trials. The algorithms were trained on the summed spikes of one half, and performance evaluated on the other half as a function of the tensor rank.

We benchmark our algorithm against both CP and Poisson GCP[2]. The latter is defined using an exponential link function for easier model comparison as well as more natural interpretation of the neural data. In plots, we refer to our method as VB-GCP (Variational Bayes GCP). All algorithms were trained with a maximum of 10000 iterations. For VB-GCP, we used $k_{0\lambda} = 100$, $\theta_{0\lambda} = 1$

---

[2]https://github.com/hugosou/vbgcp and https://www.tensortoolbox.org

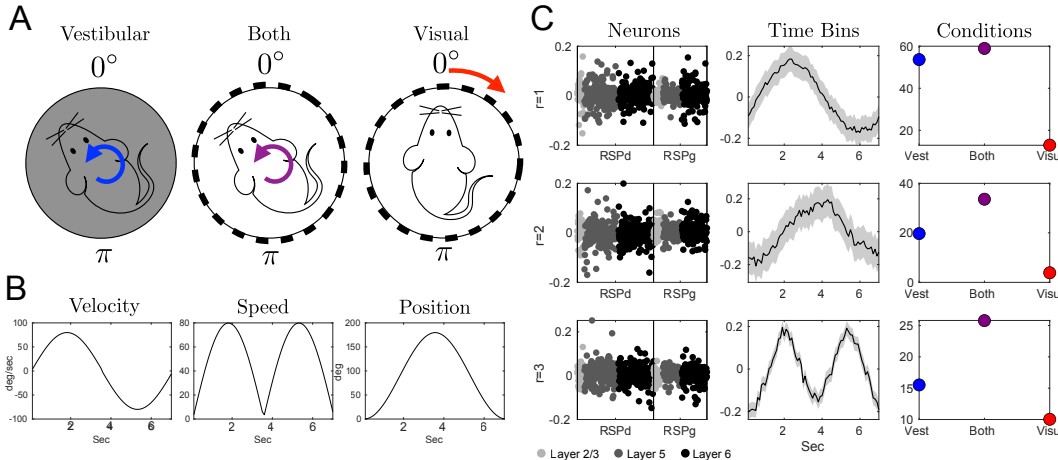

Figure 4: Neural data analysis. (A) Experimental conditions: the mouse is rotated in the dark (Vestibular), rotated in presence of a surround static visual scene (Both), or is stationary while the surrounding visual scene is being rotated (Visual). (B) Temporal profiles of the rotation stimuli under all conditions. (C) First three discovered factors using our decomposition. Grey patches account for 1 standard deviation around the mean based on variational posteriors estimates.

(see (16)). Neuron groups (see (17)) were based on the neuron recording sites (Layer and RSP division), and the offset tensor was only allowed to vary across the neuron and experimental condition dimensions.

We report testing variance explained ($VE$) and Poisson deviance explained ($DE$) as well as a similarity metric (all defined in Supplementary Section C). The latter is essentially a normalised scalar product between different the CP factors (on a scale of 0 to 1) that accounts for possible permutations of the components. Similarity is measured for factors obtained with different initializations and from different cross-validation folds. Thus it reflects both algorithmic stability and stability of the extracted structure across different trials under consistent experimental conditions.

We find that VB-GCP outperforms CP and GCP, even though $VE$ and $DE$ explicitly assess performance under the Gaussian and Poisson likelihood losses assumed by those methods (Figure 3 A-B). At low tensor rank, some of the improvement might come from the inclusion of the constrained offset tensor in the model. In particular, for a fixed rank $R$, our model includes more parameters than standard (G)CP. For a dataset of size $N \times T \times K$ (neurons $\times$ times $\times$ conditions/trials), a standard rank-$R$ CP decomposition has $R \times (N + T + K)$ parameters, to which the offset adds $N \times K$. However, VB-GCP test performance continues to climb with increased $R$, while the maximum-likelihood approaches appear to overfit even though they are applied to a model with fewer parameters (both VE and DE on training data exceed that of the VB model, but test performance falls; Figure 11 Supplementary Materials). Thus, the robustness of VB estimation plays an important role in facilitating the offset expansion of the model. While the expanded model appears valuable in the context of our experimental data, it may be omitted in other contexts without disrupting the remainder of the algorithm.

We also note that VB-GCP generates more reliable decompositions across cross-validation folds (Figure 3 C). This reliability reinforces the interpretability of the specific decompositions found, making them of particular practical value in a neuroscience context.

The Poisson distribution is a standard model of stochastic neuronal firing [21] and so we present Poisson-GCP as a relatively familiar baseline model. Our dataset displayed only moderate overdispersion (see Supplementary Materials), and so was close to the Poisson noise assumption. Furthermore, we compared model performance in terms of Poisson Deviance Explained (DE), which is the loss directly optimized by Poisson GCP. We also fit NB-GCP models but did not find notable performance improvements compared to the Poisson case (see Supplementary Section E).

Finally, we considered the interpretability of the factors found by the different models. In Supplementary Figures 12,13 and 14, for each method, we report the rank $R = 6$ decomposition which was the most similar to others across folds. Contrary to CP and GCP, our method is able to report posterior estimates of the factors, and, thanks to the offset tensor model, it directly renders patterns of modulation around condition-dependent baseline values. The discovered temporal dynamics are smoother and centered, hence easier to compare between each other and to relate to experimental variables. In Figure 4-C we show the first three inferred components (sorted by amplitude). Strikingly, and even though they are extracted in a fully unsupervised manner, they are highly reminiscent of the experimental motion variables (namely the angular velocity, position, and absolute angular speed, 4-B). Particularly, one can notice that component $r = 3$, which correlates almost perfectly with mouse absolute rotation velocity, corroborates the decoding analysis performed in [18]. In the latter, combined vestibular and visual information led to higher decoding accuracy. Here, it led to bigger modulatory effects on population activity as shown in the "conditions" panel. Both analyses thus suggest vestibulo-visual integration of rotation speed. In summary, our analysis revealed the encoding of motion variables while disentangling the contribution of different sensory modalities and brain region or layers.

## 5  Discussion

We have introduced a variational Bayesian framework for probabilistic tensor decomposition of count data, amenable and scalable to neural datasets. Our algorithm provides a principled way to estimate the rank of the data and can incorporate prior knowledge about the neural recordings (such as the neuron shape or the recording site). In practice, and although this comes at a significant computational cost compared to finely optimized blackbox tensor toolboxes [9], our algorithm leads to improved reconstruction performance, and perhaps more importantly to greater robustness compared to baseline (G)CP decompositions. That is, our method was both able to better capture variance in the dataset, and recovered decompositions that were more stable across experimental trials. Such stability is key in order to draw accurate conclusions from the multi-way analysis of neural datasets. One additional feature of our method is the introduction of the constrained offset tensor to the decomposition. In the context of the multisensory integration experiment [18] presently analyzed, it factors out the baseline firing rates of neurons and centers temporal factors, which facilitates the interpretation of the modulatory impact of experimental parameters on neural activity.

Our method builds upon previous work using PG augmentation schemes [11–13] and extends it to a Bayesian treatment of tensor decomposition. Although the PG approach was also adopted in [14], the authors used it to develop an EM algorithm for maximum a posterior point-estimates of the tensor factors, with no posterior variance. Moreover, they proposed a different way of inferring the tensor rank, relying on a multiplicative Gamma prior on the factor amplitudes. Although this scheme also allows probabilistic treatment of the rank, the focus on the amplitudes prevents the model from incorporating knowledge about specific modes of the dataset (such as neuron groups in the present paper). In contrast, we made a variational treatment of the factors possible by approximating PG KL divergences with their Gamma moment-matched counterparts in regions where we expect them to be well behaved. Nevertheless, as with many mean-field or factored variational inference procedures, it is possible that our method underestimate the true uncertainty in the model variables. Possible extensions could therefore involve low-rank approximations to the posterior covariance between factors, or the extension of linear-response methods [22] to PG-augmented models that lie outside the exponential family.

Finally, our model can cope with Poisson ($FF = 1$) and overdispersed datasets ($FF > 1$) but underdispersion ($FF < 1$) has been reported in several neural circuits like the early visual system [23], the somatosensory cortex [24], the Dorsal Premotor Cortex (after presentation of a stimulus) [20] or the auditory cortex [25]. Interestingly, in this last example, DeWeese and Zador found that single-unit spiking activity was underdispersed to the point that the majority of neurons exhibited binary behavior with few multi-spike responses. In this particular case, the Pólya-Gamma augmentation based tensor decomposition could easily be adapted by switching from a negative binomial to binomial observation model using the original model from [11]. In the other cases, neuron responses seem more complex and would require a more sophisticated extension of our model. One solution to low variability settings is to model spiking history and neuron coupling (see for example [26]), but its incorporation to the tensor factorisation framework would require further developments.

## Acknowledgments and Disclosure of Funding

We would like to thank Joaquin Rapela for early contributions to this project and Marc Deisenroth and Céline Marié for helpful comments on the manuscript. This work was funded by the Simons Foundation and the Gatsby Charitable Foundation.

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
