# Supplementary Materials: Probabilistic Tensor Decomposition of Neural Population Spiking Activity

**Hugo Soulat**
Gatsby Unit
University College London
London, W1T 4JG
hugos@gatsby.ucl.ac.uk

**Sepiedeh Keshavarzi**
Sainsbury Wellcome Centre
University College London
London, W1T 4JG
s.keshavarzi@ucl.ac.uk

**Troy W. Margrie**
Sainsbury Wellcome Centre
University College London
London, W1T 4JG
t.margrie@ucl.ac.uk

**Maneesh Sahani**
Gatsby Unit
University College London
London, W1T 4JG
maneesh@gatsby.ucl.ac.uk

## A    Details of variational updates

In this section, we derive the variational updates. The complete log joint likelihood of the augmented model can be written

$$\log L = \log P(\mathcal{X}, \mathcal{U}, \mathcal{V}, \mathcal{W}, \Lambda | \zeta, \mu, \tau, k, \theta) = \sum_{\bar{d}} \log \mathcal{P}(\mathcal{X}_{\bar{d}}, \mathcal{U}_{\bar{d}} | \mathcal{V}_{\bar{d}}, \mathcal{W}_{\bar{d}}, \zeta)$$

$$+ \sum_{\bar{d}*} \log \mathcal{N}\left(\nu_{\bar{d}*} | \mu_{\bar{d}*}, 1/\tau_{\bar{d}*}\right) + \sum_{n,i} \log \mathcal{N}\left(a_i^{(n)} | 0, \left(\Lambda_i^{(n)}\right)^{-1}\right) + \log P(\Lambda | k_\lambda, \theta_\lambda), \quad (28)$$

where the first term represents the Pólya-Gamma augmented factor defined in (12) and (13),

$$\log \mathcal{P}(\mathcal{X}_{\bar{d}}, \mathcal{U}_{\bar{d}} | \mathcal{V}_{\bar{d}}, \mathcal{W}_{\bar{d}}, \zeta) = \log p_{\mathrm{PG}}(\mathcal{U}_{\bar{d}} | \zeta + \mathcal{X}_{\bar{d}}, 0)$$

$$- \tfrac{1}{2}\mathcal{U}_{\bar{d}}(\mathcal{W}_{\bar{d}} + \mathcal{V}_{\bar{d}})^2 - \tfrac{1}{2}(\mathcal{X}_{\bar{d}} - \zeta)(\mathcal{W}_{\bar{d}} + \mathcal{V}_{\bar{d}}) + \log F(\mathcal{X}_{\bar{d}}, \zeta), \quad (29)$$

and we recall that $\mathcal{V}$ is determined by $\nu$, and $\mathcal{W} = [|A^{(1)}, \ldots, A^{(D)}|]$ with $a_i^{(n)}$ the rows of $A^{(n)}$.

The VB factored posterior on $\mathcal{Z} = \{\mathcal{U}, \mathcal{V}, \mathcal{W}, \Lambda\}$ is given by,

$$q(\mathcal{Z}) = q(\mathcal{U})q(\mathcal{V})q(\mathcal{W})q(\Lambda) = \prod_{\bar{d}} q(\mathcal{U}_{\bar{d}}) \prod_{\bar{d}*} q(\nu_{d*}) \prod_{n,i} q(a_i^{(n)}) q(\Lambda | k_\lambda, \theta_\lambda). \quad (30)$$

For each $X \in \{\mathcal{U}, \mathcal{V}, \mathcal{W}, \Lambda\}$, we represent by $q_{\neg X}$ the variational distribution marginalized over $X$. Then each VB inference update takes the form

$$q(X) \propto \exp\langle \log P(\mathcal{X}, \mathcal{Z} | \zeta, \mu, \tau, k, \theta) \rangle_{q_{\neg X}} \quad (31)$$

We note that the expectation of the PG augmentation term (29) with respect to $q(\mathcal{U})$ can be expressed as

$$\langle \log \mathcal{P}(\mathcal{X}, \mathcal{U} | \mathcal{V}, \mathcal{W}, \zeta) \rangle_{q(\mathcal{U})} =_{+C} -\frac{1}{2} \sum_{\bar{d}} \langle \mathcal{U}_{\bar{d}} \rangle \left( \mathcal{W}_{\bar{d}} + \mathcal{V}_{\bar{d}} - \frac{\mathcal{X}_{\bar{d}} - \zeta}{2\langle \mathcal{U}_{\bar{d}} \rangle} \right)^2, \quad (32)$$

and introduce the notation

$$\langle \mathcal{Y} \rangle = \frac{\mathcal{X} - \zeta}{2\langle \mathcal{U} \rangle}, \quad (33)$$

for compactness, retaining the expectation brackets to emphasise the dependence on $q(\mathcal{U})$.

35th Conference on Neural Information Processing Systems (NeurIPS 2021).

## A.1 Updates for low rank tensor $\mathcal{W}$

For $n = 1 \ldots D$, $i = 1 \ldots I_n$:

$$\langle \log L \rangle_{q_{\neg a_i^{(n)}}} =_{+C}$$
$$- \frac{1}{2} \left\langle \sum_{i=1}^{I_n} \left( a_i^{(n)} B^{(n)\mathsf{T}} - \langle \mathcal{Y}_{(n),i:}^{\mathcal{V}} \rangle \right) \mathrm{Diag}\left( \mathcal{U}_{(n),i:} \right) \left( a_i^{(n)} B^{(n)\mathsf{T}} - \langle \mathcal{Y}_{(n),i:}^{\mathcal{V}} \rangle \right)^{\mathsf{T}} \right\rangle_{q_{\neg a_i^{(n)}}}, \tag{34}$$

where $\mathcal{Y}^{\mathcal{V}} = \mathcal{V} - \langle \mathcal{V} \rangle$. Keeping only the terms in $a_i^{(n)}$, we can write the update:

$$q\left( a_i^{(n)\mathsf{T}} \right) = \mathcal{N}\left( m_i^{(n)}, \Sigma_i^{(n)} \right), \tag{35}$$

where

$$\Sigma_i^{(n)} = \left( \left\langle B^{(n)\mathsf{T}} \mathrm{Diag}\left( \langle \mathcal{U}_{(n),i:} \rangle \right) B^{(n)} \right\rangle + \left\langle \Lambda_i^{(n)} \right\rangle \right)^{-1}$$
$$m_i^{(n)} = \Sigma_i^{(n)} \left( \langle B^{(n)\mathsf{T}} \rangle \mathrm{Diag}\left( \langle \mathcal{U}_{(n),i:} \rangle \right) \right) \left( \langle \mathcal{Y}_{(n),i:} \rangle - \langle \mathcal{V}_{(n),i:} \rangle \right). \tag{36}$$

In this expression:

- $\langle \mathcal{U} \rangle$ is given in closed form using the mean of a PG distribution and (47).
- $\langle B^{(n)} \rangle$ is obtained by replacing $a_l^{(m)}$ by $m_l^{(m)}$ (for $m \neq n$) in the Khatri-Rao product.

To simplify this experession further, we write $\tilde{A}^{(n)}$ for the $I_n$ by $R^2$ matrix whose rows are: $\tilde{a}_i^{(n)} = \mathrm{Vec}\left( m_i^{(n)\mathsf{T}} m_i^{(n)} + \Sigma_i^{(n)} \right)^{\mathsf{T}}$ and $\tilde{B}^{(n)} = \bigodot_{p \neq n} \tilde{A}^{(p)}$. If $j_p$ is the index associated to the $p$-th factor in the $j$-th row $B_{j:}^{(n)}$ of the Khatri-Rao product $B^{(n)}$, then for $r_1, r_2 = 1 \ldots R$:

$$\langle B^{(n)\mathsf{T}} \mathrm{Diag}\left( \mathcal{U}_{(n),i:} \right) B^{(n)} \rangle_{r_1 r_2} = \sum_j \langle \mathcal{U}_{(n)ij} \rangle \prod_{p \neq n} \left( m_{j_p}^{(p)\mathsf{T}} m_{j_p}^{(p)} + \Sigma_{j_p}^{(p)} \right)_{r_1, r_2}, \tag{37}$$

or, using "$\circ$" to denote the Hadamard product and $\mathbb{1}_{1 \times k}$ a 1 by $k$ vector of ones,

$$\mathrm{Vec}\left[ \langle B^{(n)\mathsf{T}} \mathrm{Diag}\left( \mathcal{U}_{(n),i:} \right) B^{(n)} \rangle \right]^{\mathsf{T}} = \mathbb{1}_{1 \times (\prod I_p)} \left( \tilde{B}^{(n)} \circ \left( \mathbb{1}_{1 \times R} \otimes \langle \mathcal{U}_{(n),i:} \rangle \right) \right) \in \mathbb{R}^{1 \times R^2}.$$

The various components can then be reassembled to give the moments of $q(\mathcal{W})$:

$$\langle \mathcal{W}_{\bar{d}} \rangle = [| \langle A^{(1)} \rangle, \ldots, \langle A^{(D)} \rangle |]_{\bar{d}} \quad \text{and} \quad \langle \mathcal{W}_{\bar{d}}^2 \rangle = [| \tilde{A}^{(1)}, \ldots, \tilde{A}^{(D)} |]_{\bar{d}}.$$

### A.1.1 Updates for factor precisions $\Lambda$

By conjugacy, the precision update takes the form

$$q(\Lambda | k, \theta) \sim \mathrm{Gamma}(k_\lambda, \theta_\lambda). \tag{38}$$

**Case 1: Rank ARD.** For $r = 1 \ldots R$, if $\mathcal{D}$ indicates the dimensions sharing a precision matrix, we have

$$k_{\lambda,r} = k_{0\lambda} + \frac{1}{2} \sum_{n \in \mathcal{D}} I_n \quad \text{and} \quad \theta_{\lambda,r} = \left( \theta_{0\lambda}^{-1} + \frac{1}{2} \sum_{n \in \mathcal{D}} \sum_{i=1}^{I_n} \left\langle a_{ir}^{(n)2} \right\rangle \right)^{-1}. \tag{39}$$

**Case 2: Neuron-Group constraints.** For $r = 1 \ldots R$ and a neuron group $g \in \mathcal{G}$

$$k_{\lambda,gr} = k_{0\lambda} + \frac{1}{2} |g| \quad \text{and} \quad \theta_{\lambda,gr} = \left( \theta_{0\lambda}^{-1} + \frac{1}{2} \sum_{i \in g} \left\langle a_{ir}^{(n)2} \right\rangle \right)^{-1}. \tag{40}$$

### A.1.2 Updates for offset tensor $\mathcal{V}$

For $\bar{d}^* = (d_1^*, \ldots, d_{D_1}^*)$, we want to update $\nu_{\bar{d}*}$. Gathering the terms repeated across constrained dimensions, we get:

$$\langle \log L \rangle_{q_{\neg(\nu_{\bar{d}*})}} =_{+C} \left( -\frac{1}{2} \sum_{\bar{d}\bullet} \langle \mathcal{U}_{\bar{d}*\cup\bar{d}\bullet} \rangle \right) \nu_{\bar{d}*}^2 + \left( \sum_{\bar{d}\bullet} \langle \mathcal{U}_{\bar{d}*\cup\bar{d}\bullet} \rangle \langle \mathcal{Z}_{\bar{d}*\cup\bar{d}\bullet} \rangle \right) \nu_{\bar{d}*} . \tag{41}$$

Therefore

$$q(\nu_{\bar{d}*}) = \mathcal{N}\left( m_{\bar{d}*}, \Sigma_{\bar{d}*}, \right) , \tag{42}$$

with

$$\Sigma_{\bar{d}*} = \left( \tilde{\mathcal{U}}_{\bar{d}*} + \tau_{\bar{d}*} \right)^{-1} \quad \text{and} \quad m_{\bar{d}*} = \Sigma_{\bar{d}*} \left( \tilde{\mathcal{U}}_{\bar{d}*} \tilde{\mathcal{Y}}_{\bar{d}*} + \tau_{\bar{d}*} \mu_{\bar{d}*} \right) , \tag{43}$$

where

$$\tilde{\mathcal{U}}_{\bar{d}*} = \sum_{\bar{d}\bullet} \langle \mathcal{U}_{\bar{d}*\cup\bar{d}\bullet} \rangle \quad \text{and} \quad \tilde{\mathcal{Y}}_{\bar{d}*} = \frac{\sum_{\bar{d}\bullet} \langle \mathcal{U}_{\bar{d}*\cup\bar{d}\bullet} \rangle \left( \langle \mathcal{Y}_{\bar{d}*\cup\bar{d}\bullet} \rangle - \langle \mathcal{W}_{\bar{d}*\cup\bar{d}\bullet} \rangle \right)}{\sum_{\bar{d}\bullet} \langle \mathcal{U}_{\bar{d}*\cup\bar{d}\bullet} \rangle} . \tag{44}$$

### A.1.3 Updates for Pólya-Gamma latents $\mathcal{U}$

For $\bar{d} = (d_1, \ldots, d_D)$, we leverage (9) to rewrite:

$$\left\langle \log \left( e^{-\frac{\mathcal{U}_{\bar{d}}}{2}(\mathcal{W}_{\bar{d}}+\mathcal{V}_{\bar{d}})^2} p(\mathcal{U}_{\bar{d}}|\zeta + \mathcal{X}_{\bar{d}}, 0) \right) \right\rangle_{q_{\neg\mathcal{U}_{\bar{d}}}} = \log \left( e^{-\frac{\mathcal{U}_{\bar{d}}}{2}\Omega_{\bar{d}}} p(\mathcal{U}_{\bar{d}}|\zeta + \mathcal{X}_{\bar{d}}, 0) \right)$$
$$=_{+C} \log p(\mathcal{U}_{\bar{d}}|\zeta + \mathcal{X}_{\bar{d}}, \Omega_{\bar{d}}) , \tag{45}$$

where

$$\Omega_{\bar{d}} = \sqrt{\left\langle (\mathcal{W}_{\bar{d}} + \mathcal{V}_{\bar{d}})^2 \right\rangle} . \tag{46}$$

Thus, we find that

$$q(\mathcal{U}_{\bar{d}}) = \mathrm{PG}\left( \zeta + \mathcal{X}_{\bar{d}}, \, \Omega_{\bar{d}} \right) . \tag{47}$$

## A.2 Rewriting the free energy

The free energy (or ELBO) is given by:

$$\mathcal{F}(\zeta) = \langle \log P(\mathcal{X}, \mathcal{Z}|\zeta) \rangle_q + \mathcal{H}[q] . \tag{48}$$

Dropping terms in the log joint probability which do not depend on $\zeta$:

$$\mathcal{F}(\zeta) =_{+C} \langle \log P(\mathcal{X}|\mathcal{Z}, \zeta) \rangle_q + \langle \log P(\mathcal{U}; \zeta + \mathcal{X}, 0) \rangle_q + \mathcal{H}[q(\mathcal{U})]$$
$$=_{+C} \sum_{\bar{d}} \log \Gamma(\mathcal{X}_{\bar{d}} + \zeta)/\Gamma(\zeta) - \zeta \left( \log 2 + \langle \mathcal{W}_{\bar{d}}/2 \rangle + \langle \mathcal{V}_{\bar{d}}/2 \rangle \right) \tag{49}$$
$$- \sum_{\bar{d}} \langle \log q(\mathcal{U}_{\bar{d}}) - \log P(\mathcal{U}_{\bar{d}}; \zeta + \mathcal{X}_{\bar{d}}, 0) \rangle_{q(\mathcal{U}_{\bar{d}})} .$$

The last term contains cross entropies between prior and variational PG distributions. We then use the fact that after a variational E-step, $q(\mathcal{U}_{\bar{d}})$ is fixed, and to avoid confusion, we denote by $\hat{\zeta}$ the current estimate of the shape parameter. Recalling that we have:

$$P(\mathcal{U}_{\bar{d}}; \zeta + \mathcal{X}_{\bar{d}}, 0) \propto \cosh^{-(\zeta+\mathcal{X}_{\bar{d}})}(\Omega_{\bar{d}}/2) P(\mathcal{U}_{\bar{d}}; \zeta + \mathcal{X}_{\bar{d}}, \Omega_{\bar{d}}) , \tag{50}$$

we obtain:

$$\mathcal{F}(\zeta) =_{+C} \sum_{\bar{d}} \log \Gamma(\mathcal{X}_{\bar{d}} + \zeta)/\Gamma(\zeta) - \zeta \left( \log 2 + \langle \mathcal{W}_{\bar{d}}/2 \rangle + \langle \mathcal{V}_{\bar{d}}/2 \rangle + \log \cosh(\Omega_{\bar{d}}/2) \right)$$
$$- \sum_{\bar{d}} \mathrm{KL}\left( PG(\mathcal{X}_{\bar{d}} + \hat{\zeta}, \Omega_{\bar{d}}) || PG(\mathcal{X}_{\bar{d}} + \zeta, \Omega_{\bar{d}}) \right) . \tag{51}$$

# B   Asymptotic behaviour of the moment matching approximation

In this section we examine the behaviour of the moment-matched Gamma($\alpha$,$\beta$)-based approximation to the PG cross entropy terms in (27). We first explore the properties of the VB approximation to a PG-augmented negative binomial model, arguing that in realistic neural data settings solutions it will often converge to a parameter regime associated with low conditional Fano factors. We then leverage the Laplace transforms of the PG and gamma densities to show that expansions of the higher-order moments of both distributions are dominated by identical leading terms in this regime. This provides justification for the approximation, the empirical quality of which we then evaluate in numerical experiments.

## B.1   The behaviour of the Pólya-Gamma augmented model

Recall from (12) that PG augmentation replaces the factors associated with the observation likelihoods by terms derived from the identity

$$\frac{(e^{-v})^{\zeta}}{(1+e^{-v})^{\zeta+x}} \equiv 2^{-(\zeta+x)} e^{\left(\frac{x-\zeta}{2}\right)v} \int_0^{\infty} e^{-\frac{u}{2}v^2} p(u|\zeta+x,0) du \,, \tag{52}$$

where $p(u|\zeta+x,0)$ is a PG density. This augmented form would not be useful if our goal was to generate samples of $x$, as the implied distribution $p(x|u,v)$ is considerably more complex than the original negative binomial $p(x|v)$. Its value arises solely in inference, where the conditional likelihood on $v$, that is $p(x|v,u)$ viewed as a function of $v$, *is* simplified to a Gaussian-conjugate form.

This view also emphasises the fact that despite the appearance of the density $p(u|\zeta+x,0)$ in the definition, the implied prior on $u$ is more complex. Nonetheless, as we saw in the previous section, the joint likelihood is tractable and yields a PG variational posterior on $u$.

To understand the behaviour of the model when fit to neural data, it is helpful to consider the shape of the likelihood landscape generated by the underlying NB observation model

$$p(x|v,\zeta) = \frac{\Gamma(\zeta+x)}{x!\,\Gamma(\zeta)} \frac{e^{-\zeta v}}{(1+e^{-v})^{\zeta+x}} \,. \tag{53}$$

The likelihood is highest when the NB mean (given by $\zeta e^v$) is close to the observation $x$ and its dispersion, controlled by the Fano factor FF $= 1 + e^v$, is small. Thus, for a single observation, optimisation of the NB likelihood will drive $\zeta$ to diverge, while $v$ remains close to $\log(x/\zeta)$.

The full VB model affects this limit in two ways. First, the likelihoods associated with all the observations $\mathcal{X}$ are linked by the restricted structure of the tensor $\mathcal{W}+\mathcal{V}$, and so cannot be optimised individually. Second, the incorporation of posterior uncertainty on the tensor $\mathcal{W}+\mathcal{V}$ means that rather than optimising a single-point likelihood, we optimise the expectation of its logarithm under $q(\mathcal{W},\mathcal{V})$. However, a similar limit applies if the model provides a good fit to the data with low dispersion. That is, roughly, if the posterior is concentrated such that the $\mathcal{X}$ are dispersed around $\zeta e^{\langle\mathcal{W}+\mathcal{V}\rangle}$ with an effective FF approaching 1. Now, again, optimisation of the model will drive $\zeta$ to grow, while the posterior on $\mathcal{W}+\mathcal{V}$ concentrates to keep $\zeta e^{\langle\mathcal{W}+\mathcal{V}\rangle}$ close to $\mathcal{X}$. However, in this case the added constraints of the model introduce a natural limit to this divergence.

To see the impact on the inferred PG parameters, recall that the variational posterior on $\mathcal{U}$ is given by

$$q(\mathcal{U}_{\bar{d}}) = \text{PG}(\zeta+\mathcal{X}_{\bar{d}}, \Omega_{\bar{d}}) \quad \text{where} \quad \Omega_{\bar{d}} = \sqrt{\langle(\mathcal{W}_{\bar{d}}+\mathcal{V}_{\bar{d}})^2\rangle} \,. \tag{54}$$

Thus, in the "good fit" limit, the growth of $\zeta$ directly increases the first parameter of each PG posterior. Its impact on the second parameter is indirect. When $\zeta > \mathcal{X}_{\bar{d}}$, the corresponding tensor entry will be negative, and grow more negative with increasing $\zeta$ (thus driving FF $\to$ 1). As $\Omega_{\bar{d}}$ depends on its magnitude, we see that it too grows, but at a rate given by $\log\zeta$.[3]

How plausible is this "good fit" regime? Single neuron spike-counts over repeated trials of an experiment are often observed to be substantially overdispersed relative to Poisson [16]. However,

---

[3]Parenthetically, we note that this behaviour would be more evident in a different but broadly equivalent parametrisation of the likelihood model as $\mathcal{X}_{\bar{d}} \sim \text{NB}\big(\zeta, 1/(1+e^{(\mathcal{W}_{\bar{d}}+\mathcal{V}_{\bar{d}})-\log\zeta})\big)$.

much of this variability appears to be correlated across neurons [27] and population studies often report conditional Fano factors closer to 1 once such common variance has been captured by a latent variable model such as ours [20]. Thus, this "good fit" setting is one that we expect to be encountered in many neural data sets. For the neural data explored in this study we found a overall conditional FF of approximately 1.17.

## B.2 Moment-matched gamma distribution

Matching the first two central moments of $\mathrm{Gamma}(\alpha, \beta)$ to those of $\mathrm{PG}(\xi, \omega)$ (11) we have

$$\frac{\alpha}{\beta} = \xi \frac{\tanh(\omega/2)}{2\omega} \quad \text{and} \quad \frac{\alpha}{\beta^2} = \frac{\xi}{4\omega^3 \cosh^2(\omega/2)} \left(\sinh(\omega) - \omega\right) . \tag{55}$$

As we will see below, these forms determine the leading terms in the higher-order (non-central) moments.

## B.3 Higher-order moments

We write $f_{PG}$ and $f_G$ for the Laplace transforms of $\mathrm{PG}(\xi, \omega)$ and $\mathrm{Gamma}(\alpha, \beta)$ respectively, and recall that

$$f_{PG}(t) = \cosh^\xi(\omega/2) \cosh^{-\xi}\left(\sqrt{\frac{\omega^2/2 + t}{2}}\right) \quad \text{and} \quad f_G(t) = \left(1 + \frac{t}{\beta}\right)^{-\alpha} . \tag{56}$$

The $n$th moment of each distribution is given by the $n$th derivative with respect to $-t$ at 0 of the corresponding Laplace transform, that is $(-1)^n f^{(n)}(0)$.

For $\mathrm{Gamma}(\alpha, \beta)$, we have immediately that

$$f_G^{(n)}(0) = (-1)^n \frac{\alpha(\alpha+1)\ldots(\alpha+n-1)}{\beta^n} . \tag{57}$$

The corresponding form for the Pólya-Gamma is more complex. However, we show by induction that for all $n \in \mathbb{N}$, the derivative can be written as a sum of at most $3^n$ terms in polynomials $\{P_i^n\}$ and $\{Q_i^n\}$, indexed by tuples $i \in \{1, 2, 3\}^n$.

$$f_{PG}^{(n)}(t) = \frac{\cosh^\xi(\omega/2)}{4^n} \left(\sum_i \frac{P_i^n[\xi] \, Q_i^n[\cosh g_\omega(t), \sinh g_\omega(t)]}{\cosh^{\xi+p_i} g_\omega(t) \, g_\omega(t)^{d_i}}\right) , \tag{58}$$

where

$$g_\omega(t) = \sqrt{\frac{\omega^2/2 + t}{2}} \tag{59}$$

and the degrees of the polynomials satisfy

$$0 \le d(Q_i^n) \le d(P_i^n) = p_i \le n \le d_i . \tag{60}$$

The result follows immediately for $n = 0$ by inspection of the Laplace transform (56). Assume it holds for $n$. Then differentiating (58) with respect to $t$ and noting that $g'_\omega(t) = 1/(4g_\omega(t))$ yields:

$$
\begin{aligned}
f_{PG}^{(n+1)}(t) = \frac{\cosh^\xi(\omega/2)}{4^n} \sum_i \Bigg( & \frac{P_{i,1}^{n+1}[\xi] \, Q_{i,1}^{n+1}[\cosh g_\omega(t), \sinh g_\omega(t)]}{\cosh^{\xi+p_i+1} g_\omega(t) \, g_\omega(t)^{d_i}} \\
& + \frac{P_{i,2}^{n+1}[\xi] \, Q_{i,2}^{n+1}[\cosh g_\omega(t), \sinh g_\omega(t)]}{\cosh^{\xi+p_i} g_\omega(t) \, g_\omega(t)^{d_i}} \\
& + \frac{P_{i,3}^{n+1}[\xi] \, Q_{i,3}^{n+1}[\cosh g_\omega(t), \sinh g_\omega(t)]}{\cosh^{\xi+p_i} g_\omega(t) \, g_\omega(t)^{d_i+1}} \Bigg) \left(\frac{1}{4g_\omega(t)}\right) ,
\end{aligned}
\tag{61}
$$

where,

$$
\begin{aligned}
& P_{i,1}^{n+1}[\xi] = -P_i^n[\xi](\xi + p_i) && Q_{i,1}^{n+1} = Q_i^n[\cosh g_\omega(t), \sinh g_\omega(t)]\sinh g_\omega(t) \\
& P_{i,2}^{n+1}[\xi] = P_i^n[\xi] && Q_{i,2}^{n+1} = \frac{d}{dg_\omega} Q_i^n[\cosh g_\omega(t), \sinh g_\omega(t)] \\
& P_{i,3}^{n+1}[\xi] = -P_i^n[\xi]d_i && Q_{i,3}^{n+1} = Q_i^n .
\end{aligned}
\tag{62}
$$

Note that as $\cosh'(x) = \sinh(x)$ and vice-versa, $Q_{i,2}^{n+1}$ is of the same degree as $Q_i^n$. Thus, all the degree conditions (60) are preserved for $n+1$, completing the induction. Note in particular that induction on the first term of the expansion in (61) leads to the only term that satisfies the conditions with equality: $d(Q_{1,1,\ldots}^n) = d(P_{1,1,\ldots}^n) = p_1 = n = d_{1,1,\ldots}$. This term has the form:

$$\cosh^\xi(\omega/2)\frac{(-1)^n(\xi(\xi+1)(\xi+2)\ldots(\xi+n-1))\sinh^n g_\omega(t)}{4^n\cosh^{\xi+n} g_\omega(t)\, g_\omega(t)^n} \tag{63}$$

Evaluating (58) at $t = 0$ yields:

$$\begin{aligned}
f_{PG}^{(n)}(0) = &\frac{(-1)^n(\xi(\xi+1)(\xi+2)\ldots(\xi+n-1))\sinh^n(\omega/2)}{\cosh^n(\omega/2)\,(2\omega)^n} \\
&+ \sum_i \frac{P_i^n[\xi]Q_i^n[\cosh(\omega/2),\sinh(\omega/2)]}{4^n\cosh^{p_i}(\omega/2)\,(\omega/2)^{d_i}}\,.
\end{aligned} \tag{64}$$

where we have separated the leading term from the remainder of the sum.

## B.4 Asymptotic behaviour

We recall that the mean and variance of the NB model are given by:

$$\mathbb{E}(\mathcal{X}|\zeta,\mathcal{W},\mathcal{V}) = \zeta e^{\mathcal{W}+\mathcal{V}} \quad \text{and} \quad \mathbb{V}(\mathcal{X}|\zeta,\mathcal{W},\mathcal{V}) = \zeta e^{\mathcal{W}+\mathcal{V}}(1 + e^{\mathcal{W}+\mathcal{V}})\,, \tag{65}$$

We can therefore define:

$$\begin{aligned}
FF &= \langle 1 + e^{\mathcal{W}+\mathcal{V}}\rangle_q \\
&= 1 + e^{\langle\mathcal{W}+\mathcal{V}\rangle + \frac{1}{2}\left(\langle(\mathcal{W}+\mathcal{V})^2\rangle - \langle\mathcal{W}+\mathcal{V}\rangle^2\right)}\,.
\end{aligned} \tag{66}$$

Now consider a single $q(\mathcal{U}_{\bar{d}}) = PG(\xi,\omega)$ with $\xi = \zeta + \mathcal{X}_{\bar{d}}$ and $\omega = \Omega_{\bar{d}}$. Based on the argument of Appendix B.1 in the "good fit" limit, $\zeta \approx \mathcal{X}_{\bar{d}}/(FF-1)$ and $\langle(\mathcal{W}_{\bar{d}}+\mathcal{V}_{\bar{d}})^2\rangle = \mathcal{O}\left(\langle\mathcal{W}_{\bar{d}}+\mathcal{V}_{\bar{d}}\rangle^2\right)$, so:

$$\begin{aligned}
\omega &= -\langle\mathcal{W}_{\bar{d}}+\mathcal{V}_{\bar{d}}\rangle\sqrt{1 + \frac{\langle(\mathcal{W}_{\bar{d}}+\mathcal{V}_{\bar{d}})^2\rangle - \langle\mathcal{W}_{\bar{d}}+\mathcal{V}_{\bar{d}}\rangle^2}{\langle\mathcal{W}_{\bar{d}}+\mathcal{V}_{\bar{d}}\rangle^2}} \\
&= -\langle\mathcal{W}_{\bar{d}}+\mathcal{V}_{\bar{d}}\rangle + \frac{\langle(\mathcal{W}_{\bar{d}}+\mathcal{V}_{\bar{d}})^2\rangle - \langle\mathcal{W}_{\bar{d}}+\mathcal{V}_{\bar{d}}\rangle^2}{\langle\mathcal{W}_{\bar{d}}+\mathcal{V}_{\bar{d}}\rangle} + \mathcal{O}\left(\frac{(\langle(\mathcal{W}_{\bar{d}}+\mathcal{V}_{\bar{d}})^2\rangle - \langle\mathcal{W}_{\bar{d}}+\mathcal{V}_{\bar{d}}\rangle^2)^2}{\langle\mathcal{W}_{\bar{d}}+\mathcal{V}_{\bar{d}}\rangle^3}\right) \\
&= -\log(FF-1) + \frac{1}{2}\left(\langle(\mathcal{W}_{\bar{d}}+\mathcal{V}_{\bar{d}})^2\rangle - \langle\mathcal{W}_{\bar{d}}+\mathcal{V}_{\bar{d}}\rangle^2\right) + \mathcal{O}\left(\frac{\langle(\mathcal{W}_{\bar{d}}+\mathcal{V}_{\bar{d}})^2\rangle - \langle\mathcal{W}_{\bar{d}}+\mathcal{V}_{\bar{d}}\rangle^2}{\langle\mathcal{W}_{\bar{d}}+\mathcal{V}_{\bar{d}}\rangle}\right) \\
&= -\log(FF-1)\left(1 + \underset{FF\to 1}{\mathcal{O}}(1)\right)
\end{aligned} \tag{67}$$

As a conclusion, although both $\omega$ and $\xi$ will grow as $FF \to 1$, we have:

$$\omega = \log\xi\left(1 + \underset{FF\to 1}{\mathcal{O}}(1)\right)\,, \tag{68}$$

which we will now use to look at the higher moments of $PG(\omega,\xi)$ and Gamma($\alpha,\beta$). For the latter, from (57), we have:

$$\begin{aligned}
m_G^n = (-1)^n f_G^{(n)} &= \frac{\alpha(\alpha+1)\ldots(\alpha+n-1)}{\beta^n} \\
&= \left(\frac{\alpha}{\beta}\right)^n\left(1 + \mathcal{O}\left(n^2\cdot\frac{\alpha}{\beta^2}\cdot\left(\frac{\alpha}{\beta}\right)^{-2}\right)\right) \\
&= \left(\frac{\xi}{2\omega}\right)^n\left(\tanh^n(\omega/2) + \mathcal{O}\left(\frac{n^2}{\xi\omega}\frac{\sinh(\omega)-\omega}{\sinh^2(\omega/2)}\right)\right) \\
&= \left(\frac{\xi}{2\omega}\right)^n + \underset{FF\to 1}{\mathcal{O}}\left(\frac{\xi^{n-1}}{\omega^n}(1 + \xi e^{-\omega})\right)\,,
\end{aligned} \tag{69}$$

giving a leading term of $(\xi/2\omega)^n$ as $FF \to 1$.

For the PG distribution (64) gives (recalling that $d(Q_i^n) \le d(P_i^n) = p_i \le n \le d_i$ with all equalities holding only for the leading term)

$$
\begin{aligned}
m_{PG}^n = (-1)^n f_{PG}^{(n)} &= (-1)^n \sum_i \frac{P_i^n[\xi] Q_i^n[\cosh(\omega/2), \sinh(\omega/2)]}{4^n \cosh^{p_i}(\omega/2) \, (\omega/2)^{d_i}} \\
&= \left(\frac{\xi}{2\omega}\right)^n (\tanh^n(\omega/2)) + \mathcal{O}\left(\frac{n^2 3^n}{\omega}\left(\frac{\xi}{2\omega}\right)^{n-1}\right) \\
&= \left(\frac{\xi}{2\omega}\right)^n + \underset{FF\to 1}{\mathcal{O}}\left(\frac{\xi^{n-1}}{\omega^n}(1 + \xi e^{-\omega})\right),
\end{aligned}
\tag{70}
$$

Thus, we see that both sets of higher-order moments are equivalent in the leading terms of their expansions.

### B.5   Numerical evaluation of the moment matching approach.

In this section, we report a range of numerical examples and experiments to evaluate the moment matching approximation and its asymptotic behavior.

We first compared the accuracy with which the Pólya-Gamma density could be approximated using moment-matched gamma, generalised gamma, inverse gamma and normal distributions. We exploited the convolutional property of the PG distribution to obtain numerical density estimates. Each sample-based density was estimated with at least 100000 draws using the bayesreg toolbox [28]. Visually, the gamma distribution provides the best approximation, following the sampled PG density closely for $\xi \ge 10$ (Fig. 5).

We quantified the match using the normalised $L_2$ discrepancy, obtained analytically from the PG, gamma and normal characteristic functions; that is, for $X \in \{G, IG, N\}$,

$$
d(f_{PG}, f_X) = \frac{2\|p_{PG} - p_X\|_{L_2}^2}{\|p_{PG}\|_{L_2}^2 + \|p_X\|_{L_2}^2} .
\tag{71}
$$

This $L_2$ discrepancy was consistently smallest for the gamma-based approximation (Fig. 6).

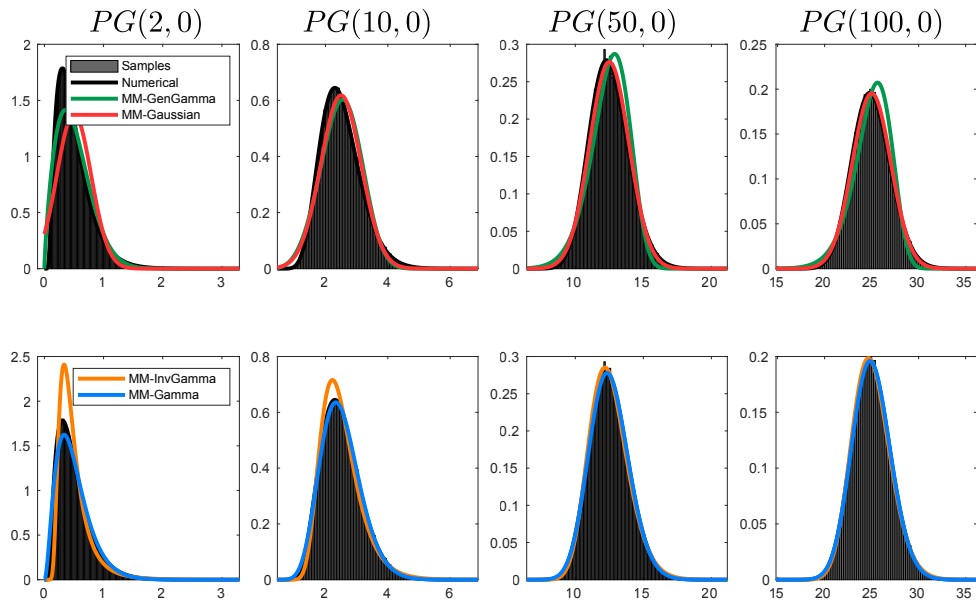

Figure 5: Densities of gamma, inverse gamma, normal and generalised gamma distributions moment-matched to PG$(\xi, 0)$ for different values of $\xi$ (In the generalized gamma case, the first three moments were matched).

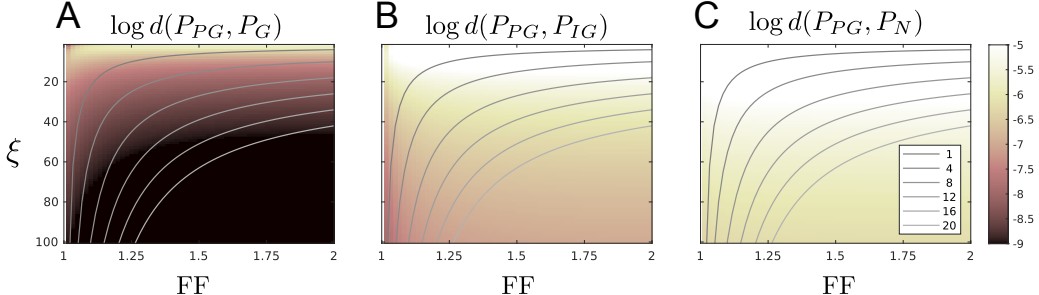

Figure 6: Normalized $L_2$ norm between $p_{PG}$ and $p_X$ for $X \in \{G, IG, N\}$ as a function of the Fano factor FF $= 1 + e^{-\omega}$ and $\xi$. The lines indicate the expected relationship between $\xi$ and FF for different numbers of observed spikes.

Finally, we compared numerical estimates of the KL divergences between PG distributions in non diverging regions, to the closed form divergence between the moment-matched gamma approximations (Fig. 7; comparisons are anchored on $PG(50, 1)$). We found the match to be good, particularly when the divergence was relatively small.

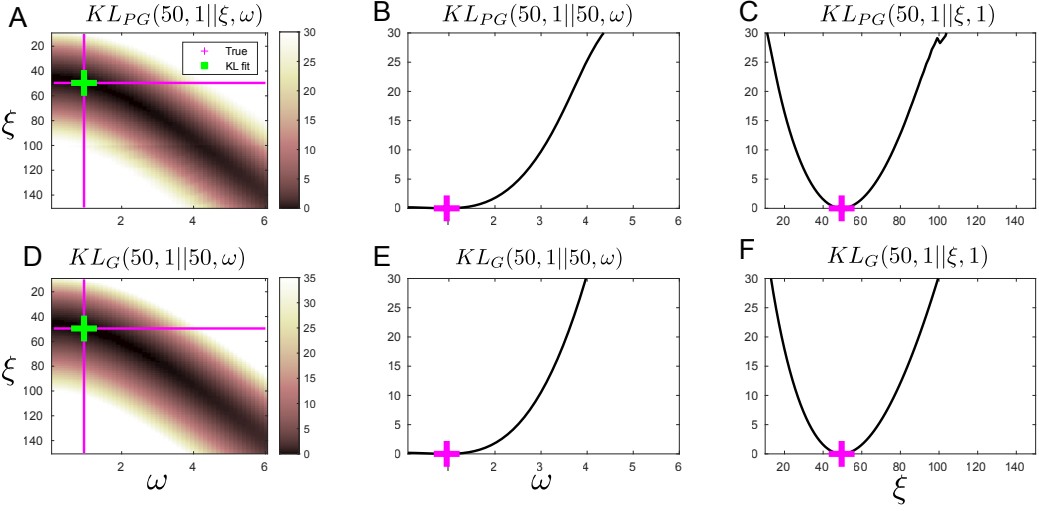

Figure 7: KL divergence between PG distributions $PG(50, 1)$ and $PG(\xi, \omega)$ obtained by numerical estimation (A-C) and moment matching mean and variance of Gamma distribution (D-E). Crosses indicate the minimum of the KL.

## C   Metrics

Performances are reported in terms of Variance Explained ($VE$), deviance explained ($DE$) and a similarity metric, which we define here. If $\mathcal{X}_0$ is the average spike count of the dataset across all conditions, neurons and time points, and $\hat{\mathcal{X}}$ a reconstructed tensor from the decompositions estimates, the $VE$ and Poisson $DE$ are given by

$$VE = 1 - \frac{\sum_{\bar{d}}(\hat{\mathcal{X}}_{\bar{d}} - \mathcal{X}_{\bar{d}})^2}{\sum_{\bar{d}}(\mathcal{X}_0 - \mathcal{X}_{\bar{d}})^2} \quad \text{and} \quad DE = 1 - \frac{\sum_{\bar{d}}(\mathcal{X}_{\bar{d}}\log \mathcal{X}_{\bar{d}}/\hat{\mathcal{X}}_{\bar{d}} + \hat{\mathcal{X}}_{\bar{d}} - \mathcal{X}_{\bar{d}})}{\sum_{\bar{d}}(\mathcal{X}_{\bar{d}}\log \mathcal{X}_{\bar{d}}/\mathcal{X}_0 + \mathcal{X}_0 - \mathcal{X}_{\bar{d}})} . \tag{72}$$

Although we assessed similarities across both cross-validation folds and initializations, we used the same metric as employed in Ref. [7]. Given two CP [9] decompositions $A = [[A^{(1)}, \dots, A^{(D)}]]$ and $V = [[V^{(1)}, \dots, V^{(D)}]]$, we normalize each column $A^{(n)}_{0,:r} = A^{(n)}_{:r}/||A^{(n)}_{:r}||_{L_2}$ and $V^{(n)}_{0,:r} = V^{(n)}_{:r}/||V^{(n)}_{:r}||_{L_2}$ and gather the normalization constants across modes in $\gamma^A_r$ and $\gamma^V_r$. In essence, the similarity metric is a generalized scalar product ($\in [0,1]$) between factors which accounts for possible permutations ($\sigma \in \sigma_R$) of the components:

$$S = \max_{\sigma \in \sigma_R} \sum_{r=1}^{R} \left( \frac{|\gamma^A_r + \gamma^V_{\sigma(r)}| - |\gamma^A_r - \gamma^V_{\sigma(r)}|}{|\gamma^A + \gamma^V_{\sigma(r)}| + |\gamma^A_r - \gamma^V_{\sigma(r)}|} \right) \prod_{n=1}^{D} A^{(n)\mathsf{T}}_{0,:r} V^{(n)}_{0,:\sigma(r)} , \tag{73}$$

We used an efficient search over permutation by leveraging Munkres (Hungarian) algorithm for the linear assignment problem.

# D Qualitative Experiments

Figures 8 to 10 show the results of two qualitative experiments described in Section 4.1. Briefly, in both cases, we generated a 5 dimensional dataset (size $100 \times 70 \times 3 \times 5 \times 4$) using a generative NB model with shape $\zeta = 80$ and $R = 4$. Each CP-component loads only on a subsets of "neurons". We assumed that the subsets were known and used them to define neuron-group priors (section 3.1).

In the first experiment, we added an offset tensor that varies across the first and third dimensions and treated only 1/4 of the full tensor $\mathcal{X}$ as observed. In this stitching setting, each "neuron" is observed in a single "experimental session", which is accounted by the last dimension (not plotted Figure 8) of the observed tensor.

The algorithm converged over about 4000 iterations (Fig. 8) and was able to retrieve the dataset shape parameters and infer both the tensor factors and rank in a fully probabilistic manner (Fig. 9 left). When the algorithm was run from the same initial values, but without ARD or neuron group constraints it inaccurately identified additional components with similar amplitudes to the true factors (Fig. 9 right).

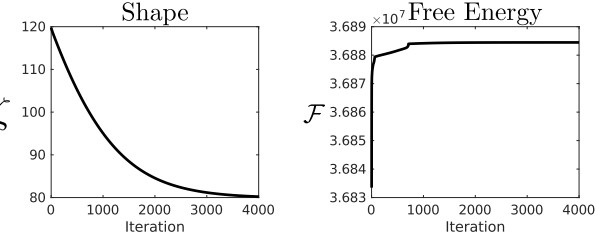

Figure 8: Evolution of the shape parameter (Left) and approximate free energy (Right) across variational EM iterations for probabilistic decomposition of the count tensor from Section 4.1. Inference used ARD and knowledge about the "neuron" groups.

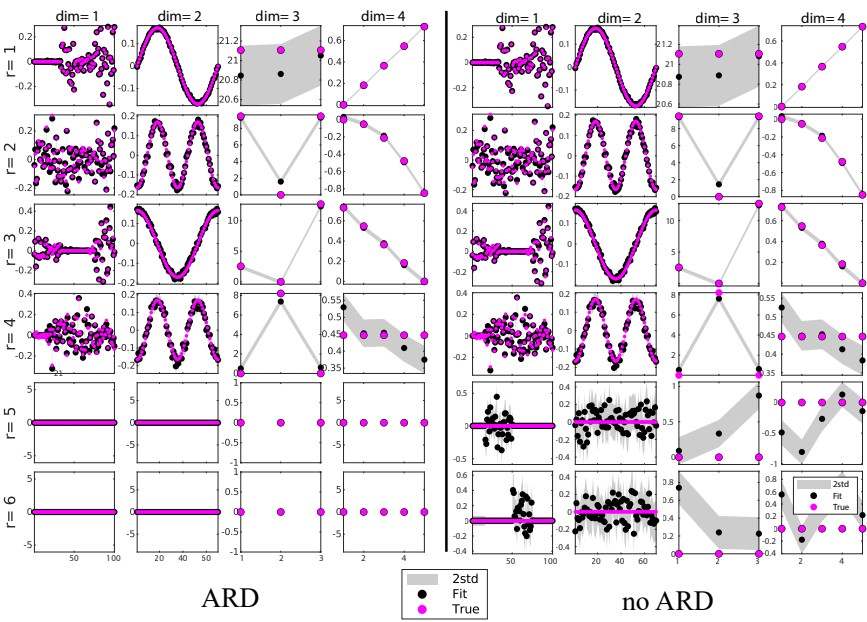

Figure 9: Probabilistic decomposition of the count tensor from Section 4.1. Ground truth generating values are in pink, estimates in black; grey shading represents 2 standard deviations around the mean based on variational posteriors. Each row represents an estimated CP component, with each column reflecting one dimension. We compare inference results using ARD and knowledge about the "neuron" groups (left) or not (right).

In the second experiment, we illustrate how standard GCP decompositions are prone to misidentify the dataset rank and how this can affect the decomposition itself. We removed the offset and tested our method against GCP with NB observation model [10] for a range of shape parameter $\zeta$ (GCP-NB required an additional hyperparameter selection step). We report the error (defined as the squared Euclidean distance between the reconstructed tensor $\hat{\mathcal{X}}$ and the true $\mathbb{E}(\mathcal{X}|\zeta,\mathcal{W})$) and the similarities between decompositions obtained with 20 random initializations for each algorithm and each putative rank. Results are reported Figure 10. The decompositions that are plotted maximize the similarity metric across the initialization sets. With standard GCP method, it is not straightforward to balance goodness of fit with robustness. Although the results suggest that all models require at least 4 components to accurately reconstruct the simulated dataset, one might need to fix an arbitrary thresholds to select the tensor rank, which can lead to incorrect rank selection or bad decomposition (Figure 10 top and bottom right).

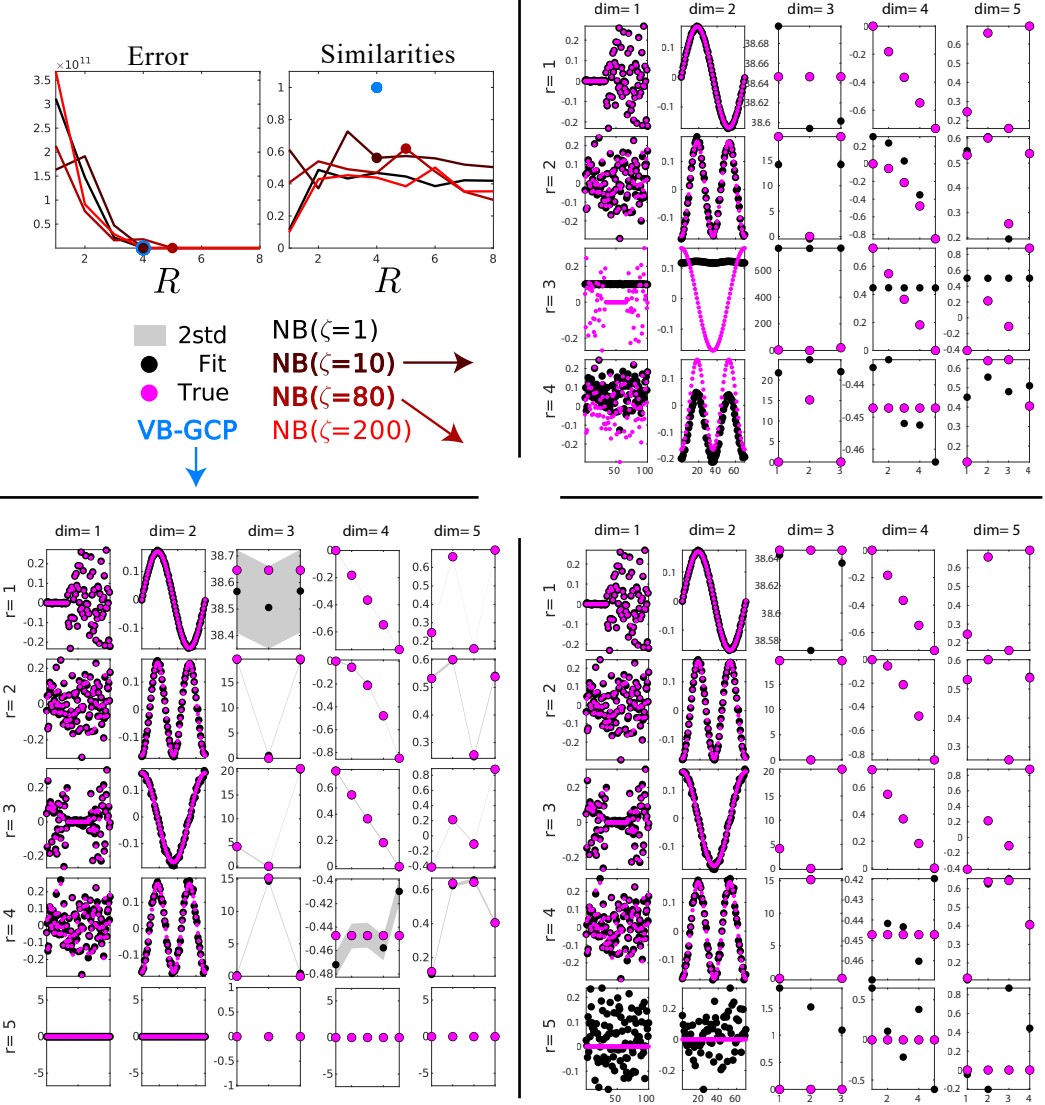

Figure 10: Goodness of Fit and Similarities (Top Left) are often balanced to select the rank of an empirical tensor decomposition. As a consequence, standard GCP (Right: top and bottom) decomposition might suffer from rank (or other hyperparameter) misidentification. Our probabilistic method (VB-GCP) provides a principled way to select the tensor rank, incorporate prior knowledge about the data and it estimates factors posterior. The decompositions plotted here maximize the similarity metric across 20 random initializations for a given rank.

# E Spike Recordings

In this section, we detail the results of the benchmark analysis described in Section 4.2. Training and testing results are plotted in Figure 11 and Figures 12 to 14 show the full VB-GCP (ours), CP and GCP decompositions described in Section 4.2. As mentioned in the main text, we also tested NB observation models with exponential link functions. Hong et al. [10] worked with a fixed shape parameter for the NB likelihood. As the most appropriate parameter value for our data was unknown, we ran a gridsearch using 50 shape parameters ranging from 1 to 100 and looked at the best models in terms of DE, Robustness (as assessed by the similarity metric) or NB likelihood. NB-GCP turned out to perform little or no better than Poisson GCP.

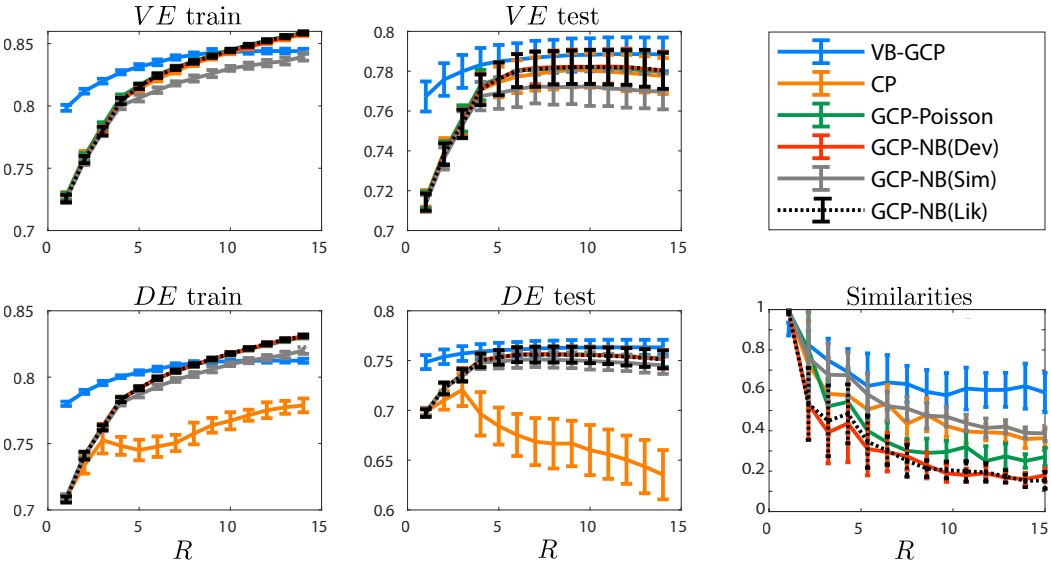

Figure 11: Train and Test variance and deviance explained for VB-GCP (ours), CP, Poisson and NB GCP as a function of the model Tensor rank. NB models plotted here were selected by maximizing similarities (grey), test $DE$ (red) or test likelihood (black).

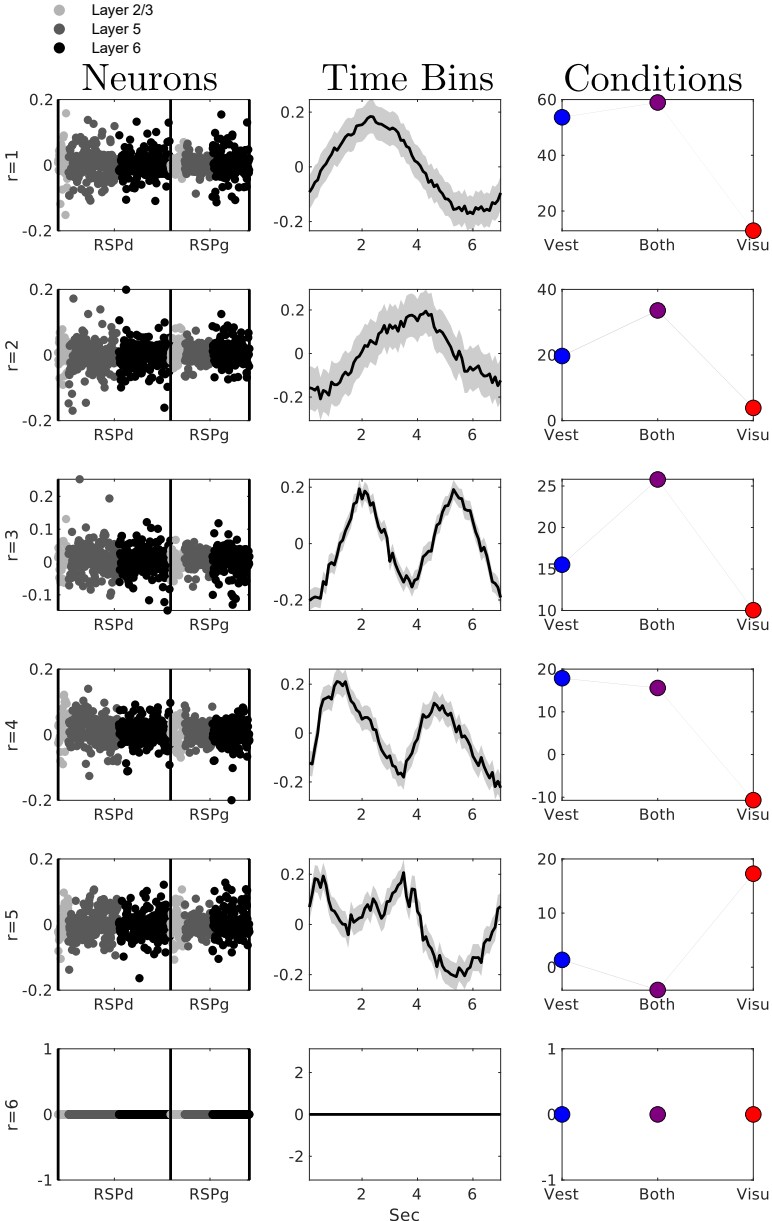

Figure 12: Full inferred factors obtained with our probabilistic decomposition. Grey patches represent 1 standard deviation around the mean based on variational posteriors.

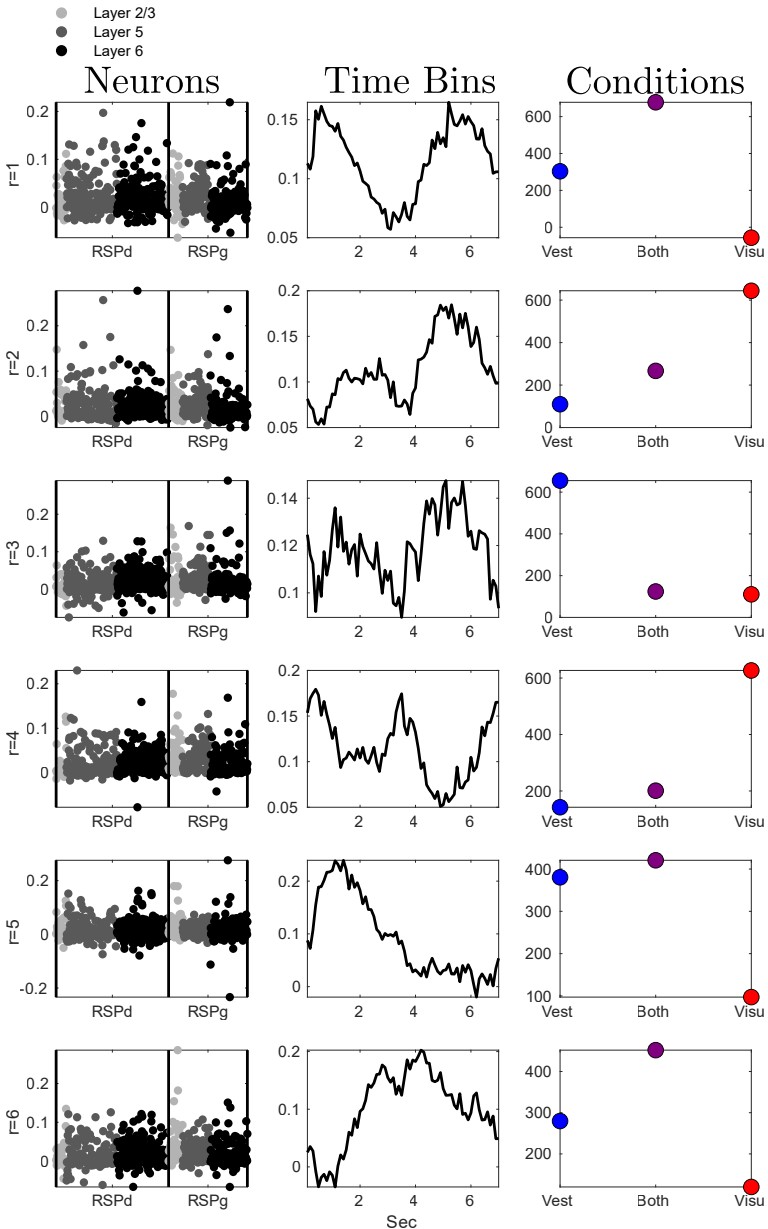

Figure 13: Full inferred factors obtained with standard CP decomposition.

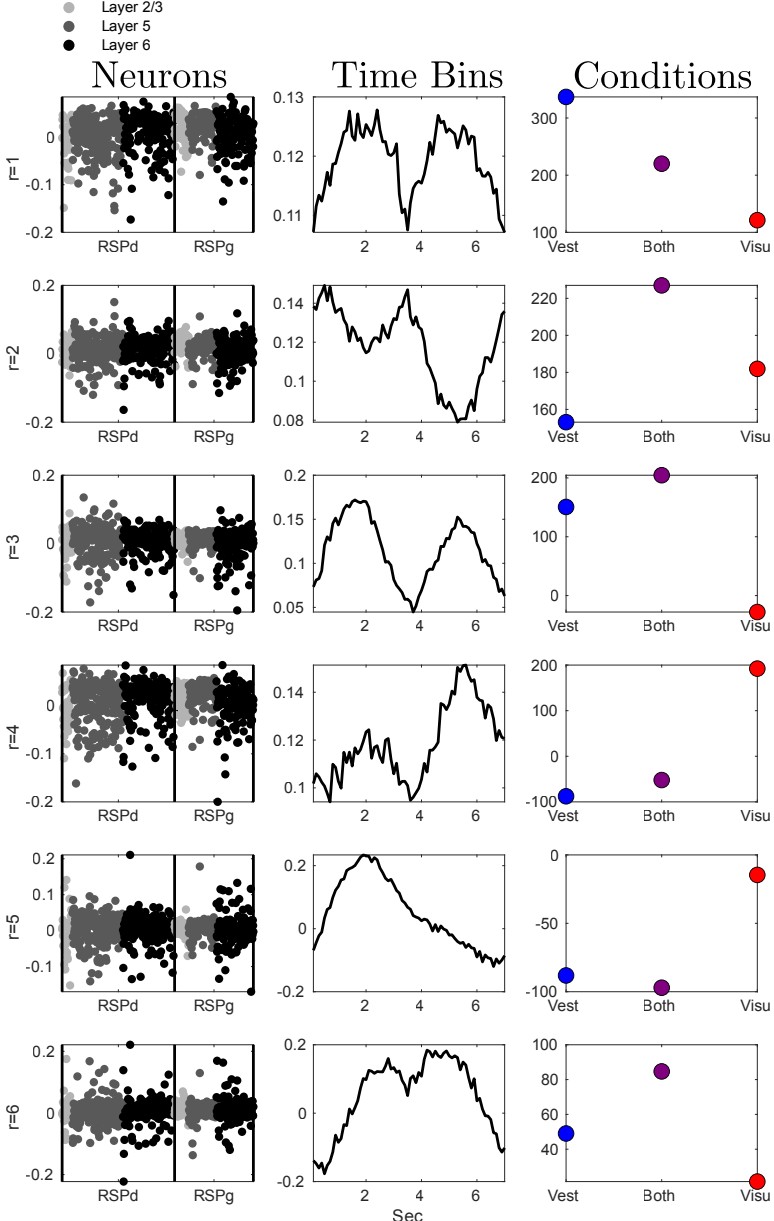

Figure 14: Full inferred factors obtained with standard GCP decomposition.