# OpenReview forum: "Probabilistic Tensor Decomposition of Neural Population Spiking Activity"
_NeurIPS.cc/2021/Conference — NeurIPS 2021 Spotlight_

### Official Review · Reviewer_dzgw · 2021-07-15

**Rating:** 8
**Confidence:** 3

**Summary:**

In the paper "Probabilistic Tensor Decomposition of Neural Population Spiking Activity" the authors propose a probabilistic model for canonical polyadic tensor decomposition and apply it to neural data. The model assumes negative binomial observations and is used with variational Bayesian inference. The method is evaluated on synthetic data and on multisensory integration activity recorded from mice.

**Limitations And Societal Impact:**

The author discuss limitations with regard to uncertainty estimates in the Discussion. The limitations paragraph could have been more extensive as the authors did not use all of the available space. I would have welcomed a discussion on the negative binomial assumption given that sometimes neuronal data have a fano factor less than 1.

**Main Review:**

Originality:
As noted by the authors, Rai et al. IJCAI 2015 used the Polya-Gamma approach as well in the tensor decomposition setting but did not achieve posterior variance estimates. For all I know, the proposed method enabling to probabilistically determine the rank and estimate variance in the CP decomposition setting is novel. The method makes use of a negative binomial observation model which is appropriate for many neuroscience datasets.

Quality:
The quality of the work is outstanding. The method is derived in a sensible and rigorous way. However, I did not check all equations.
The evaluation includes both synthetic data with known ground truth as well as data recorded from mice illustrating the utility of the approach on real data as well as yielding interesting results with regard to multisensory integration factors.
I found it surprising that the proposed method outperformed the baseline (G)CP decompositions on variance explained and Poisson deviance explained even though these are what the loss functions of the baseline models optimize for. In the Discussion the authors speculate that this is due to the constrained offset tensor. Couldn't this be easily tested with a variant of the proposed method without the offset tensor?

Clarity:
The manuscript is very well written. The work is well motivated and presented with a useful illustration (Figure 1). All figures are informative and can be understood easily.

Significance:
Given datasets of growing sizes in neuroscience, tensor decompositions are promising tools to find easy to interpret patterns in these datasets. A probabilistic model that can be used to quantify uncertainty and to determine the rank automatically is very useful in this context.

**Time Spent Reviewing:**

5

---

> ### Author Response · Authors · 2021-08-10
> **Extended discussion on the negative binomial assumption, clarification on model performances and additional test.**
>
> We thank reviewer 'dzgw' for investing time in thoroughly studying our manuscript and for their thoughtful feedback.
>
> One comment concerned the surprisingly better test performance of our model on variance and deviance explained compared to baseline decompositions (Canonical Polyadic and Generalized Canonical Polyadic, (G)CP) which directly maximise those metrics. In fact, as the tensor rank (R) increases, performance on training data of the baseline models exceeds the Bayesian approach, as might be expected (Fig 11 Supplementary Material).  Thus, the improvement appears to arise from resistance to over-fitting often seen with the (variational) Bayesian approach. This, along with the increased stability of solutions across cross-validation folds, supports the inference that the Bayesian method identifies something closer to the “true” factors.
>
>  Second, as suggested, we removed the offset tensor from our model, and although we found results closer to baseline at low R, the model still generalises better than (G)CP baselines on test data for larger R thanks to the automatic relevance determination procedure and the prior put on the neuron loading factors.
>
> Moreover, reviewer 'dzgw' rightfully mentioned the limitation of the negative binomial assumption given that sometimes, neuronal data have a fano factor (FF) smaller than 1. Although our model can cope with Poisson (FF=1) and overdispersed datasets (FF>1), it is true that underdispersion (FF<1) has been reported in several neural circuits. Kara et al. Neuron (2000) discovered that neuron spike variability was smaller than that of Poisson in the early visual system, especially for retinal cells. Churchland, Yu et. al Nature Neuroscience (2010) showed that neuron variability decreases after the presentation of a stimulus in many experimental settings, notably, FF in Dorsal Premotor Cortex could plunge below 1. Finally, DeWeese and Zador NIPS (2002) looked at multi and isolated single-unit spiking activity in auditory cortex and found that the activity of the latter was underdispersed to the point that the majority of neurons exhibited binary behavior with few multi-spike responses.
>
>  Interestingly, in this last example, our Polya-Gamma augmentation based tensor decomposition could easily be adapted by switching from a negative binomial to binomial observation model (see [12]). Nevertheless, in the other cases, neuron responses seem more complex and would require a more sophisticated extension of our model. One solution to low variability settings might be to model spiking history and neuron coupling (see for example Truccolo et al. J Neurophysiol (2005)), but incorporating it into the tensor factorisation framework through the negative binomial distribution would require extensive further development. In the present paper we would plan to acknowledge the observation of low FF in some cases and speculate as above.

---

> > ### Comment · Reviewer_dzgw · 2021-08-29
> > **Thank you for the response**
> >
> > I thank the authors for their insightful explanations in response to my comments. These confirm my initial rating.

---

### Official Review · Reviewer_1JXG · 2021-07-16

**Rating:** 8
**Confidence:** 3

**Summary:**

The paper presents a probabilistic tensor decomposition model for count data with applications to neural recordings. A variational inference method using Polya-Gamma augmentation is derived to allow feasible model fitting on large neural datasets, and some theoretical results of the asymptotic behavior of the variational approximation of the PG augmentation are shown. The utility of the tensor decomposition approach is demonstrated using recordings from rodents during a sensory integration experiment.

**Limitations And Societal Impact:**

Yes.

**Main Review:**

This proposed model is a novel approach to tensor decomposition that has strong potential to be widely applicable in neuroscience (using spike train data) than standard tensor decomposition models, which use squared error instead of a likelihood on counts.
-	Additionally, it allows for a full Bayesian treatment with good use of the Polya-Gamma augmentation in variational inference, rather than slower sampling. Advantages of this approach are demonstrated by, for example, performing rank selection using an ARD prior.

The presentation is generally clear, and appropriate comparisons to existing methods are given.
However, a couple points could improve the clarity of the results and utility of the approach:
-	The real data example did not include much discussion about what the tensor composition showed. Although comparisons to CP and GCP decompositions are provided in the supplementary, it’s not immediately clear to the reader what is gained in this case.
-	For comparing the GCP and CP to the probabilistic model (e.g., fig 3), it would be helpful to include the number of parameters in each model. I think this would help mainly because of the offset tensor included in the probabilistic model.



**Time Spent Reviewing:**

2

---

> ### Author Response · Authors · 2021-08-10
> **Results and technical clarifications**
>
> We are thankful to reviewer '1JXG' for their feedback. The comments and additional explanations requested will be incorporated into the paper.
>
> We recall that our model decomposes an observed dataset into a rank-R tensor and a constrained offset tensor. For a given R, our model therefore includes more parameters than (Generalized) Canonical Polyadic decomposition ((G)CP). As suggested, the number of parameters will be detailed in the revised version of the manuscript: for a dataset of size N x T x K (neurons x times x trials), a standard rank-R CP decomposition has R*(N+T+K) parameters, to which our model adds N*K. Yet, we stress out that this offset is an optional modeling choice, especially relevant to our experimental paradigm. Furthermore, the variance and deviance explained measure shown are on held-out data and so the added parameters did not lead to overfitting. In additional experiments without the offset the Bayesian approach still offered better fit and robustness than (G)CP at all values of R, though the advantage was much less at small R.
>
> Then, from our benchmark analysis, we compare typical tensor decompositions (Figs 12-14). Here reviewer '1JXG' considers that the gain from our method could be clarified. First, we emphasize that the Bayesian approach yields the highest similarity metric performance. Therefore, results from our algorithm (Fig 12) are the most stable across experimental trials. Second, we provide posterior variance estimates on the temporal factors. Last, the discovered temporal dynamics are smoother and centered, hence more easy to compare and to relate to experimental variables. For example, inspection of GCP component r=1 and r=3 (Fig 14 Supplementary material) hints towards neural modulation due to absolute angular velocity (see Fig 4B), but modulation amplitudes reveals that r = 1 (0.11 to 0.13 [a.u]) is mostly constant across time compared to r=3 (0.05 to 0.16 [a.u]).
>
> To provide and support a more extensive interpretation of the factors discovered (Figs 3 and 12) will require further analysis, experiments and controls, however we can add some informal observations. Particularly, one can notice that component r=3, which correlates almost perfectly with mouse absolute rotation velocity, corroborates the decoding analysis performed in [18] (on the same dataset). In the latter, combined vestibular and visual information led to higher decoding accuracy. Here, it led to bigger modulatory effects on population activity as shown in the "conditions" panel. Both analyses thus suggest vestibulo-visual integration of rotation speed.

---

> > ### Comment · Reviewer_1JXG · 2021-09-01
> > **Thanks for the response**
> >
> > I thank the authors for answering my questions. This response - and the detailed responses to the other reviews - confirm my positive rating.

---

### Official Review · Reviewer_kXjW · 2021-07-17

**Rating:** 7
**Confidence:** 3

**Summary:**

This manuscript details the inference of a probabilistic dimensionality reduction technique that is applicable to non-Gaussian data, for example spike counts. Tensor decomposition methods have been applied to neuroscience data in the past, and this paper extends this formulation to frequently seen spike data. The authors introduce a variational Bayes approach towards a Polya-Gamma augmentation of the model with negative binomial observations.

**Limitations And Societal Impact:**

Yes

**Main Review:**

The paper is very thoroughly written and presented. The model confers many advantages and the authors have presented convincing results. The model successfully recovers model parameters on simulated data while performing automatic relevance determination. On neural spiking data, the model class offers clear gains in variance explained and Poisson deviance explained in held-out data for R<4, when compared against canonical polyadic (CP) decomposition and generalized canonical polyadic (GCP) decomposition.

It would help to show the results using the CP and GCP comparison methods on simulation data to better motivate the proposed method.

It would be good to comment on the comparison between CP, GCP and the proposed method for R>=4 for the neural spiking data. Figure 3 contains the metrics for a subset of the values of R; it would be great to fill this out if possible, or provide an explanation of why these values of R were chosen.

Important: please include a detailed explanation of why GCP was chosen with the Poisson observation model as opposed to the negative binomial observation model.

Please detail out GCP before the first time it is mentioned in the manuscript, as well as include a brief description of the method.

The authors mention Polya-Gamma and its acronym PG multiple times – it would be helpful to define PG once the first time they mention it, and then stick to the acronym.


**Time Spent Reviewing:**

2

---

> ### Author Response · Authors · 2021-08-10
> **Additional explanations and results concerning our baseline choices, simulations and benchmark analysis.**
>
> We thank reviewer 'kXjW' for their time and valuable comments. We hereby address them with discussions that will be incorporated to our manuscript.
>
> The generalisation of Canonical Polyadic (CP) decomposition to non-Gaussian loss was mentioned briefly on line 32, although without introducing the GCP abbreviation. We will make this explicit and add more description as suggested. Standard CP is usually cast as a square loss minimization problem, and so it implicitly assumes that data are normally distributed. Spike counts violate this assumption but alternated-gradient-based optimization methods can be adapted to non-Gaussian likelihood functions. Although it has not been used in neuroscience to our knowledge, Hong et al. [10] introduced a general framework for Generalized CP (GCP) decomposition that can be applied to Poisson or Negative Binomial (NB) distributed datasets.
>
> Because Poisson models provide “an extremely useful approximation of stochastic neuronal firing” (p.25 Dayan and Abbott 2001) they are widely used in system neuroscience and so, we believed that this model would provide a relatable baseline for neuroscientists.  In fact, we found that our dataset only presented moderate overdispersion (see Supplementary line 463: average conditional Fano-Factor of 1.17). Furthermore we compared model performance in terms of Poisson Deviance Explained (DE), which is the loss directly optimized by Poisson GCP, making it the most appropriate baseline for this measure (NB deviance would depend on an unknown shape parameter). Finally, NB-GCP turned out to perform little or no better than Poisson GCP. Hong et al. [10] worked with a fixed shape parameter for the NB likelihood. As the most appropriate parameter value for our data was unknown, we ran a gridsearch using 50 shape parameters ranging from 1 to 100 and looked at the best models in terms of DE, Robustness (as assessed by the similarity metric) or NB likelihood but did not find notable performance improvements compared to (G)CP. Overall, we believe that the Poisson-GCP provides a strong and straightforward enough baseline to include in the main text, and we propose to include selected NB models in Fig 11 (Supplementary Materials).
>
> We observed "clear gains’' in DE in held-out data for tensor rank R<4 but higher order GCP decompositions seem to approach (from below) the performance of our model. However, we also observed that the consistency of solutions found by GCP across cross-validation folds fell significantly as R increased (Fig 3C) severely limiting the interpretability of the decomposition, key for practical implementation in neuroscience.
>
> We believe the argument for adopting the (variational) Bayesian approach goes beyond deviance-based assessment.  In particular, our model (i) provides posterior variance estimates on recovered factors; (ii) implements automatic relevance determination (ARD) for tensor rank; and (iii) makes it possible to incorporate prior knowledge about the recordings. Nevertheless, to further motivate our method, we propose to illustrate how ambiguous tensor rank selection can be. We adapted the simulations described in 4.1 to not include the offset nor missing data. On the one hand, we selected the rank of CGP-NB (with the true shape parameter) decomposition using (a) a goodness of fit metric (squared distance to the true simulated mean) and (b) the similarity metric previously discussed. On the other hand, we used our method with ARD and fixed R. While GCP-NB decompositions vary depending on the selection criteria, our algorithm provides a principled way to select parameters and better retrieves simulated factors.
>
> Benchmark results were summarised Fig 3 as a function of the tensor rank (R). We selected and tested our model for "a subset of values of R”  for clarity but, as suggested, those gaps will be filled out (the intervening points fall close to the curves shown). Finally, we thank reviewer 'kXjW' for noticing the repeated definition of the Polya-Gamma (PG) abbreviation.

---

> > ### Comment · Reviewer_kXjW · 2021-08-31
> > **Response**
> >
> > I thank the authors for clarifying their modeling and baseline choices, and I am looking forward to seeing the amendments discussed in the manuscript. Based on the authors' comments, I will improve my initial score of the paper.

---

### Decision · Program_Chairs · 2021-09-27

**Decision:**

Accept (Spotlight)

**Comment:**

The authors develop tensor decompositions for count data, like neural spike trains, as well as a variational inference approach based on the Pólya-gamma augmentation. The reviewers and I agree that the paper is novel, well-presented, and of broad interest to the computational and statistical neuroscience community at NeurIPS.